# Altered dendritic spine function and integration in a mouse model of fragile X syndrome

Sam A. Booker [1,2,3,4], Aleksander P.F. Domanski [1,7], Owen R. Dando [1,2,3,4,5], Adam D. Jackson[1,2,3,4], John T.R. Isaac[6,8], Giles E. Hardingham[1,2,3,4,5], David J.A. Wyllie [1,2,3,4] & Peter C. Kind[1,2,3,4]

Cellular and circuit hyperexcitability are core features of fragile X syndrome and related autism spectrum disorder models. However, the cellular and synaptic bases of this hyper-excitability have proved elusive. We report in a mouse model of fragile X syndrome, gluta-mate uncaging onto individual dendritic spines yields stronger single-spine excitation than wild-type, with more silent spines. Furthermore, fewer spines are required to trigger an action potential with near-simultaneous uncaging at multiple spines. This is, in part, from increased dendritic gain due to increased intrinsic excitability, resulting from reduced hyperpolarization-activated currents, and increased NMDA receptor signaling. Using super-resolution micro-scopy we detect no change in dendritic spine morphology, indicating no structure-function relationship at this age. However, ultrastructural analysis shows a 3-fold increase in multiply-innervated spines, accounting for the increased single-spine glutamate currents. Thus, loss of FMRP causes abnormal synaptogenesis, leading to large numbers of poly-synaptic spines despite normal spine morphology, thus explaining the synaptic perturbations underlying circuit hyperexcitability.

[1] Centre for Discovery Brain Sciences, University of Edinburgh, Hugh Robson Building, George Square, Edinburgh EH8 9XD, UK. [2] Patrick Wild Centre, University of Edinburgh, Hugh Robson Building, George Square, Edinburgh EH8 9XD, UK. [3] Simons Initiative for the Developing Brain, University of Edinburgh, Hugh Robson Building, George Square, Edinburgh EH8 9XD, UK. [4] Centre for Brain Development and Repair, NCBS, GKVK Campus, Bangalore 560065, India. [5] UK Dementia Research Institute, University of Edinburgh, Chancellor's Buildings, Little France, Edinburgh EH16 4SB, UK. [6] Developmental Synaptic Plasticity Section, NINDS, NIH, Bethesda, MD 20892, USA. [7] Present address: School of Physiology, Pharmacology & Neuroscience, University of Bristol, Bristol, UK. [8] Present address: Janssen Neuroscience, J&J London Innovation Centre, One Chapel Place, London W1G 0B, UK. Correspondence and requests for materials should be addressed to D.J.A.W. (email: David.J.A.Wyllie@ed.ac.uk) or to P.C.K. (email: P.Kind@ed.ac.uk)

Cell and circuit hyperexcitability have long been hypothesized to underlie many core symptoms of fragile X syndrome (FXS) and autism spectrum disorders more generally, which include sensory hypersensitivity, seizures and irritability[1]. The fundamental role of cellular excitability in circuit function raises the possibility that alterations in neuronal intrinsic physiology may underlie a range of functional endophenotypes in FXS. Despite this potential link, few studies have examined the combined synaptic, dendritic, and cellular mechanisms that lead to generation of neuronal hyperexcitability during early postnatal development.

Many cellular properties are known to regulate neuronal excitability, such as neuronal morphology, intrinsic physiology, synaptic transmission and plasticity. In FXS, a central hypothesis is that glutamatergic signalling at dendritic spines is impaired[2,3] concomitant with changes to intrinsic cellular excitability[4]. The first major alteration described was a change in dendritic spine density and morphology[3,5], however, this observation was not apparent when examined at the nanoscale using super-resolution imaging methods[6], despite an increase in synapse and spine density in the neocortex[7–9]. Notwithstanding, no study has yet observed a change in synaptic event frequency that would be predicted by a change in spine or synapse density. This has important implications for our understanding of the synaptic aetiology of FXS, as many of the current theories are reliant on altered synaptic function[10,11].

The rodent somatosensory cortex (S1) is well characterised in terms of its processing of tactile inputs, which, in the case of the barrel cortex arise from the whiskers on the facepad via relay synapses in the brainstem and ventrobasal thalamus[12]. The thalamic inputs arrive predominantly onto layer 4 stellate cells (L4 SCs) which integrate this information within L4, then project to L2/3 and L6. Furthermore, L4 SCs undergo a well-described critical period for synaptic plasticity, which closes at postnatal day 7–8 (P7–8). For these reasons, L4 of S1 provides a well-described reductionist system to examine sensory processing[13,14]. Indeed, hyperexcitability has been observed within S1 of $Fmr1^{-/y}$ mice, due in part to changes in intrinsic neuronal excitability, axonal morphology, and synaptic connectivity, which together result in increased network excitability[15–17]. The finding that the critical period for thalamocortical synaptic plasticity is delayed in $Fmr1^{-/y}$ mice compared with wild type (WT) gave a suggestion as to how cellular and circuit deficits may arise[18]. How this delay in synapse development delay affects dendritic spine function is not known. Furthermore, no study has directly examined how dendrites integrate synaptic inputs in the absence of FMRP, despite the fact that dendritic integration plays a key role in regulating cellular excitability[19–21]. Of particular relevance are findings that HCN channel expression is altered, leading to changes in intrinsic physiology and dendritic integration[16,17,22]. Here, we directly test whether there is a functional relationship between dendritic spine function, intrinsic neuronal physiology, HCN channel function, dendritic integration, and ultimately neuronal output. To address this question, we use an integrative approach that combines whole-cell patch-clamp recording from neurons in S1 at P10–14 with 2-photon glutamate uncaging, post hoc stimulated emission-depletion (STED) microscopy, and serial block-face scanning-electron microscopy.

## Results

### Larger single dendritic spine currents in $Fmr1^{-/y}$ L4 SCs.
To first assess the function of identified dendritic spines in $Fmr1^{-/y}$ mice, we performed single-spine 2-photon glutamate uncaging. Whole-cell patch-clamp recordings were performed from L4 SCs in voltage clamp with a Cs-gluconate based intracellular solution

containing a fluorescent dye (Alexafluor488, 100 μM) and biocytin to allow on-line and post hoc visualization of dendritic spines. Following filling, we performed 2-photon uncaging of Rubi-glutamate (Rubi-Glu) to elicit uncaging excitatory post-synaptic currents (uEPSCs; Fig. 1a). From both the concentration- and power–response relationships (Supplementary Fig. 1A, B), we determined that 300 μM [Rubi-Glu] and 80–100 mW laser power (λ780 nm) were optimal to produce saturating uEPSCs at −70 mV. Analysis of the spatial properties of Rubi-Glu uncaging confirmed that the optimal position for photolysis was 0–1 μm from the edge of the spine head (Supplementary Fig. 1C), and the resulting uEPSCs were blocked with CNQX, confirming that they were produced by AMPA receptors (AMPARs, Supplementary Fig. 1D). We also found no difference in spine distance from cell soma and uEPSC rise or decay time and amplitude suggesting equal space clamp of the neurons across the dendritic distances examined (Supplementary Fig. 1F-H). All details of statistical tests performed can be found in Supplementary Table 1.

Comparison between genotypes revealed that the single-spine uEPSCs in WT mice had an amplitude of $6.9 \pm 0.4$ pA ($n = 17$ mice), while $Fmr1^{-/y}$ mice ($n = 14$ mice) showed a larger uEPSC amplitude of $9.8 \pm 0.5$ pA (d.f.: 4, 5; $\chi^2 = 8.26$; $p = 0.004$; LMM, Fig. 1 and Supplementary Fig. 2), indicating that spines in $Fmr1^{-/y}$ mice are enriched for AMPAR-mediated currents (Fig. 1b, c). This difference appeared to be due to a greater population of uEPSCs at $Fmr1^{-/y}$ spines with amplitudes over 10 pA (Fig. 1b). As expected from larger underlying currents, the single-spine uncaging excitatory post-synaptic potential (uEPSP) was also larger in $Fmr1^{-/y}$ mice ($0.73 \pm 0.12$ mV, $n = 10$ mice), when compared with WT littermates ($0.47 \pm 0.06$ mV, $n = 16$ mice; d.f.: 24; $t = 2.09$; $p = 0.046$; T-test; Fig. 1d). In a subset of dendritic spines we observed no AMPAR current at −70 mV, however, a large NMDA receptor (NMDAR) current was present at + 40 mV, indicating the presence of silent dendritic spines (Fig. 1e). Quantification of the silent spines revealed an occurrence of $17.6 \pm 3.5\%$ in $Fmr1^{-/y}$ mice ($n = 13$ mice), almost threefold higher than in WT mice ($6.4 \pm 1.6\%$, $n = 17$ mice; d.f.: 27; $t = 3.1$; $p = 0.005$; T-test; Fig. 1f). When measured across all spines, the NMDA/AMPA ratio was significantly elevated as both a population average (d.f.: 1, 331; $F = 37.36$; $p < 0.0001$; F-test; Fig. 1g) and also as a spine average with $Fmr1^{-/y}$ mice having a ratio of $1.26 \pm 0.05$ ($n = 117$ spines) and WT of $0.97 \pm 0.03$ ($n = 194$ spines; $\chi^2 = 6.27$ $p = 0.012$, LMM, Fig. 1h and Supplementary Fig. 3).

Given that the majority of L4 SC dendritic spines are formed by cortico-cortical synapses in WT mice[23], and therefore likely comprise the majority of uncaged spines, we next asked whether synapses formed between L4 SCs had larger EPSC amplitudes by performing paired recordings between synaptically coupled neurons (Fig. 2). As previously described in 2-week-old mice[16], we observed a low connectivity between L4 SCs in $Fmr1^{-/y}$ mice of 14.8%, that is significantly lower than that of WT mice which had a connectivity of 33.6% ($p = 0.015$, Fisher's exact test, Fig. 2c). Despite this reduced connectivity, there was no difference in either failure rate (d.f.: 41; $t = 0.25$; $p = 0.80$; GLMM; Fig. 2d) or unitary EPSC amplitude (d.f.: 41; $t = 1.53$, $p = 0.15$; LMM; Fig. 2e), suggesting that synaptic strength is unchanged at the majority of synapses in $Fmr1^{-/y}$ mice.

### $Fmr1^{-/y}$ spines have typical morphology but more synapses.
The inclusion of biocytin within the internal solution allowed post hoc visualisation of the recorded neurons, following fixation and re-sectioning. We next performed correlated stimulated emission-depletion (STED) imaging of the same dendritic spines we had uncaged upon (Fig. 3a–e). Measurement of nanoscale spine

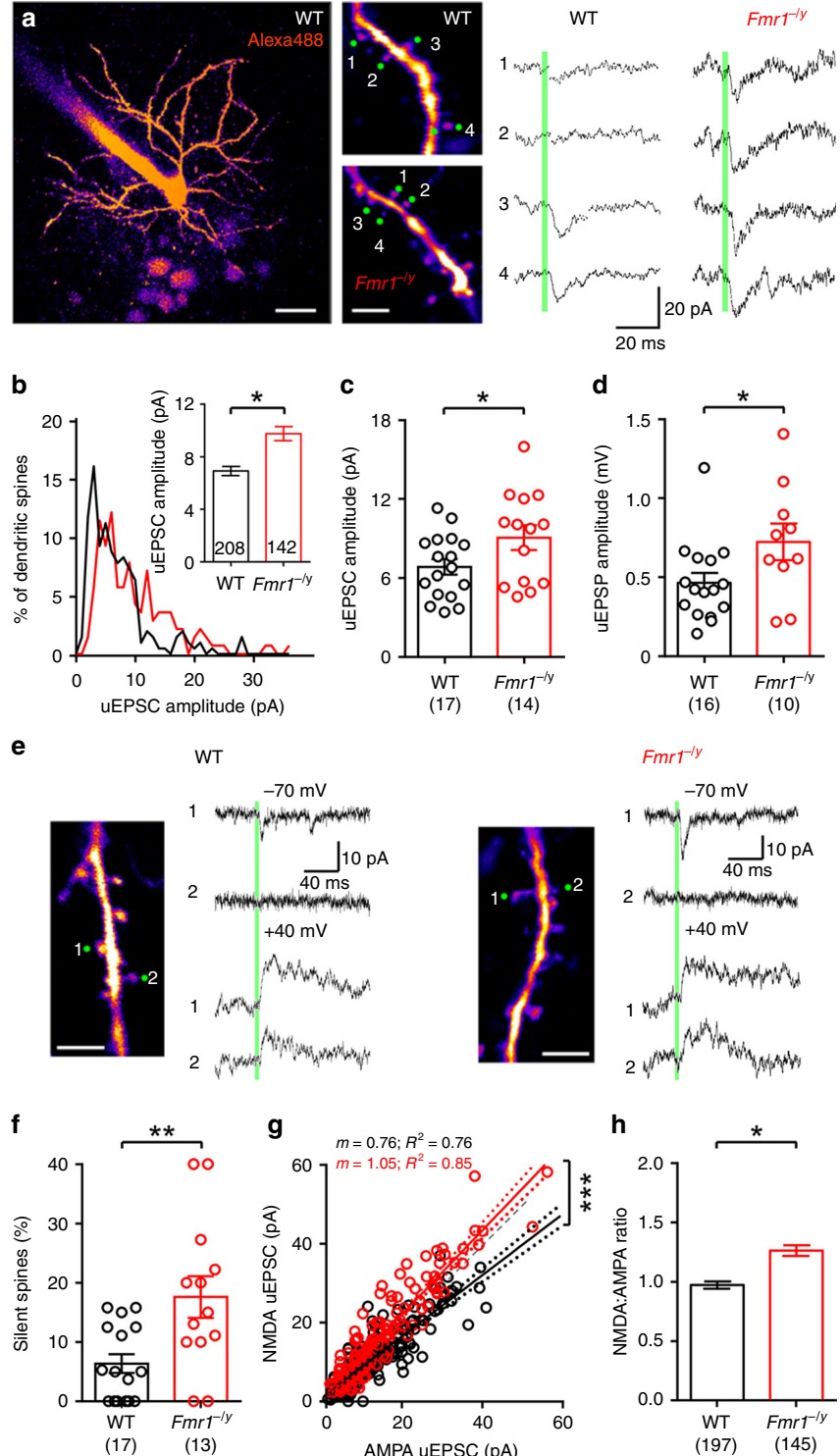

**Fig. 1** L4 SC dendritic spines have larger uEPSCs with more silent synapses in *Fmr1*[−/y] mice. **a** 2-photon image of a L4 SC (left) with selected spines and AMPAR uEPSCs from WT and *Fmr1*[−/y] mice. Scale bars: 20 μm (left), 5 μm (right). **b** Single-spine uEPSCs from WT (black) and *Fmr1*[−/y] (red) mice shown as a histogram, with spine average shown (inset). Note that spines with no AMPA response, silent spines have not been included. **c** Animal average uEPSC amplitudes, excluding silent spines. Number of animals tested shown in parenthesis. **d** Animal average of uEPSP amplitudes. **e** AMPAR (upper) and NMDAR (lower) uEPSCs, illustrating silent spines. Scale: 5 μm. **f** Incidence of silent spines in WT and *Fmr1*[−/y] mice. **g** AMPAR and NMDAR uEPSCs for all spines, with NMDA/AMPA ratio (WT: 0.76 ± 0.03; *Fmr1*[−/y]; 1.05 ± 0.04; d.f.: 1, 331; $F = 37.4$; $p < 0.0001$; F-test). **h** Average NMDA/AMPA ratio plotted for all spines. Statistics shown: *$p < 0.05$, **$p < 0.01$, from LMM (**b**, **d**, **h**), unpaired $t$-test (**c**, **f**) and sum-of-least-squares F-test (**g**). Plots of individual spine data for panels **c** (inset) and **h** can be found in Supplementary Fig. 4. All data are shown as mean ± SEM and source data for all plots are provided as a Source Data file

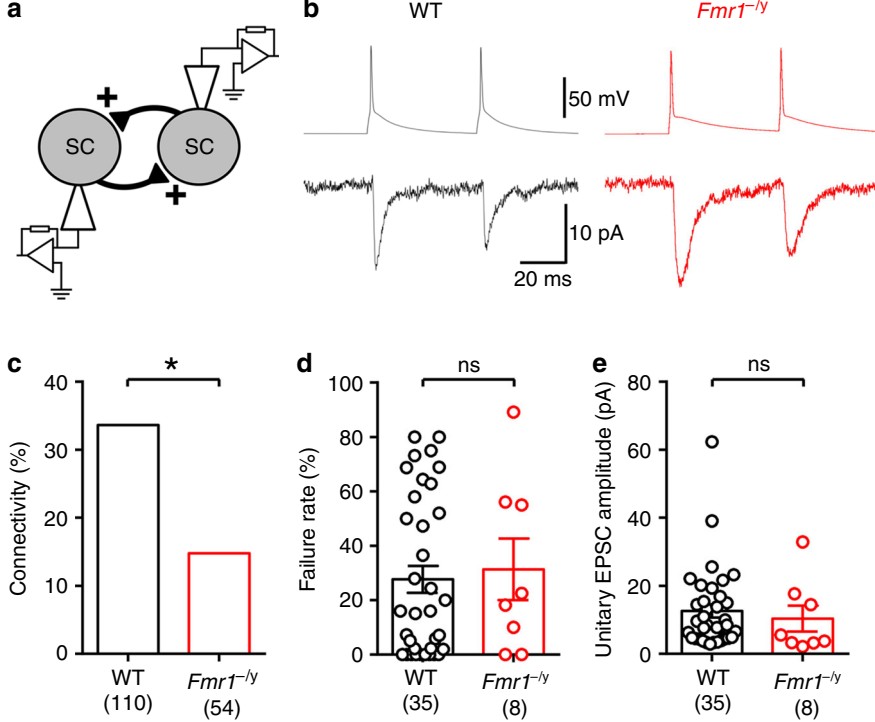

**Fig. 2** Typical EPSC amplitude at unitary connections between L4 SCs. **a** Schematic paired recordings between synaptically coupled L4 SCs.
**b** Representative presynaptic action potentials (top) produced unitary EPSCs in the second L4 SC (lower), from WT (black) and $Fmr1^{-/y}$ (red) mice.
**c** Synaptic connectivity is reduced between L4 SCs in the $Fmr1^{-/y}$ mouse (d.f.: 162; $p = 0.015$; Fisher's exact test; 110 pairs from 13 mice for WT mice and
54 pairs from 7 mice in $Fmr1^{-/y}$ mice were tested. **d** Failure rate was not different between genotypes when a connection was present. **e** Unitary EPSC
amplitudes from L4 SC synapses were not different between genotypes. Statistics shown: ns $p > 0.05$, $*p < 0.05$ from Fisher's exact test (**c**) and LMM
(**d**, **e**). All data are shown as mean ± SEM and source data for all plots are provided as a Source Data file

morphology revealed that there was no difference in either spine head width (Fig. 3b), nor neck length (Fig. 3d), between WT ($n = 6$ mice) and $Fmr1^{-/y}$ ($n = 4$ mice) mice. Consistent with earlier findings[24], we observed a weak positive correlation with spine head width and EPSC amplitude in WT mice ($7.8 ± 3.8$ pA/μm, $R^2 = 0.06$, $F = 4.3$, $p = 0.042$, F-test), which was not different to that of $Fmr1^{-/y}$ mice ($F = 0.02$, $p = 0.89$, sum-of-squares F-test; Fig. 3c). We observed no correlation with spine neck length and EPSC amplitude (Fig. 3e). To confirm that uncaging itself did not result in spine remodelling, we also measured spines from non-uncaged dendrites on filled neurons. Spine density itself was not different between genotypes (Fig. 3f), nor were head width (Fig. 3g, h) and neck length (Fig. 3i, j), in agreement with previous findings from L5 of S1 and CA1 of the hippocampus[6].

Given the strengthening of dendritic spines, but no change in unitary EPSC amplitude or spine morphology, we next asked whether the ultrastructure of dendritic spines was altered. To achieve this, we used serial block-face scanning-electron microscopy in L4 of S1 from mice perfusion fixed at P14. In serial stacks (50 nm sections; Fig. 4) we identified Type-1 asymmetric synapses on dendritic spines, based on the presence an electron dense post-synaptic density (PSD) opposing an axon bouton containing round vesicles. Following 3-dimensional reconstruction, we identified a subset of dendritic spines that contained more than one PSD, which were each contacted by an independent presynaptic axon bouton (Fig. 4a, b), and henceforth referred to as multi-innervated spines (MIS). These MIS were present in both genotypes, however, the incidence in $Fmr1^{-/y}$ mice was $20.5 ± 1.6\%$ of all spines ($n = 7$ mice), approximately threefold higher than in WT littermates ($7.2 ± 1.5\%$ of spines, $n = 3$ mice, d.f.: 8; $t = 4.9$; $p = 0.001$; T-test; Fig. 4c), which is

similar to that observed in organotypic hippocampal cultures from WT mice[25].

The presence of higher numbers of MIS in $Fmr1^{-/y}$ mice, and larger single spines uEPSCs, despite a similar density of spines and similar dendritic morphologies[26], would suggest an increased number of synapses for each L4 SC. The conventional method to assess such a change in synapse number is to perform miniature EPSC (mEPSC) recordings (Fig. 5a). AMPAR mEPSCs recorded at $-70$ mV in $Fmr1^{-/y}$ mice were very similar to WT in both amplitude (d.f.: 46; $U = 245$; $p = 0.28$; Mann–Whitney test) and frequency (d.f.: 46; $U = 240$; $p = 0.24$; Mann–Whitney test; Fig. 5b). NMDAR mEPSCs, recorded at $+40$ mV in the presence of CNQX, also had very similar amplitudes (d.f.: 17; $U = 37$; $p = 0.59$; Mann–Whitney test). However, $Fmr1^{-/y}$ mice showed a 54% increase in NMDAR mEPSC frequency compared with WT mice (d.f.: 17; $U = 18$; $p = 0.03$; Mann–Whitney test; Fig. 5c). These data indicate that while AMPAR-containing synapses number and strength are unaltered in $Fmr1^{-/y}$ mice, they possess ~50% more NMDAR containing synapses.

**$Fmr1^{-/y}$ L4 SCs are hyperexcitable due to lower HCN currents**. While these observed changes in synaptic properties reveal differences in dendritic spine function, alone they do not reveal how neurons integrate excitatory inputs leading to hyperexcitability. Dendritic spines act as spatiotemporal filters whose summation is dependent upon synaptic receptor content[21] and intrinsic membrane properties[20,27], the latter of which contributes to the cable properties of dendrites[28]. To explore the effect of altered synaptic properties on dendritic integration in $Fmr1^{-/y}$ SCs, we next measured the intrinsic excitability of L4 SCs by assessing their

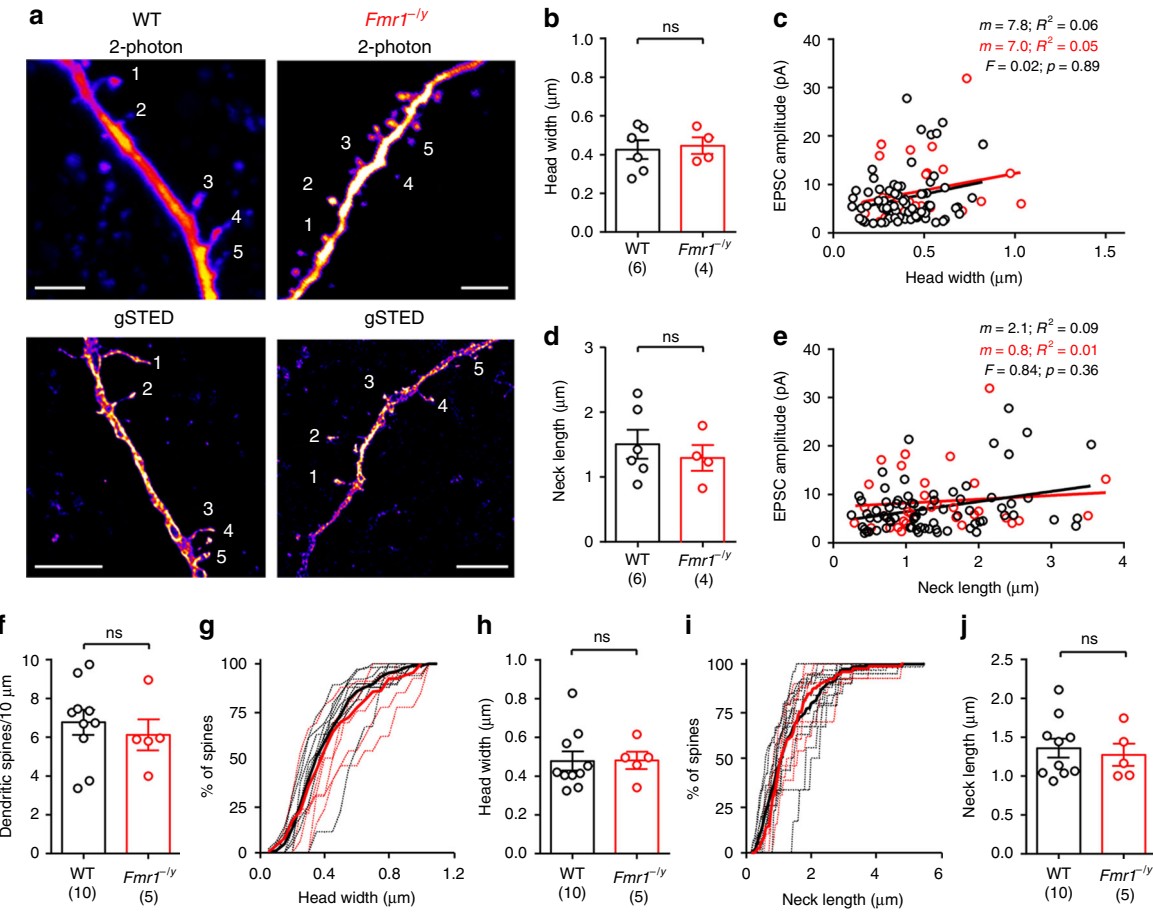

**Fig. 3** Dendritic spines show no difference in nanoscale morphology, or structure–function relationship. **a** Dendrites from WT (left) and *Fmr1*−/y (right) mice under 2-photon microscopy (top), then post hoc STED imaging (bottom). Scale bar: 5 μm. **b** Average spine head width in WT (black) and *Fmr1*−/y (red) mice (WT: 0.43 ± 0.05; *Fmr1*−/y; 0.45 ± 0.04; d.f.: 8; $t = 0.29$; $p = 0.78$, T-test). Number of mice is indicated. **c** Comparison of spine head width and uEPSC amplitude (comparing slope: d.f.: 1, 100; $F = 0.02$; $p = 0.89$). WT spines showed a positive correlation (d.f.: 70, $F = 4.27$, $p = 0.042$, F-test). **d** Average spine neck length (WT: 1.52 ± 0.22; *Fmr1*−/y; 1.31 ± 0.20; d.f.: 8; $t = 0.66$; $p = 0.53$, T-test). **e** Comparison of spine neck-width and uEPSC amplitude (Slope: WT: 2.1 ± 0.8; *Fmr1*−/y; 0.8 ± 1.4; d.f.: 1, 101; $F = 0.84$; $p = 0.36$; F-test). **f** Spine density on L4 SCs (WT: 6.8 ± 0.7 spines/10 μm; *Fmr1*−/y: 6.1 ± 0.80 spines/10 μm; d.f.: 13; $p = 0.60$; $p = 0.56$; T-test). **g** Distribution of non-uncaged spine head-widths, as an average of all mice (bold) and individual mice (dashed). **h** Average head width of non-uncaged spines (WT: 0.48 ± 0.05 μm; *Fmr1*−/y: 0.48 ± 0.04 μm; d.f.: 13; $U = 20.0$; $p = 0.59$; Mann–Whitney U-test). **i** Distribution of spine neck length of non-uncaged spines. **j** Average of spine neck length in non-uncaged spines (WT: 1.36 ± 0.12 μm; *Fmr1*−/y: 1.27 ± 0.14 μm; d.f.: 13; $U = 20.0$; $p = 0.55$; Mann–Whitney U-test). Statistics shown: ns $p > 0.05$ from unpaired t-test (**b**, **d**, **f**, **h**, **j**) and sum-of-least-squares F-test (**c**, **e**). All data are shown as mean ± SEM and source data for all plots are provided as a Source Data file

response to hyperpolarising and depolarising current injections (Fig. 6a, b). In *Fmr1*−/y mice, L4 SC input resistance ($R_I$) was increased compared with WT mice, as measured from the steady-state current–voltage relationship (Interaction: d.f.: 5, 230; $F = 7.03$; $p < 0.0001$; two-way RM ANOVA Fig. 6c) and smallest current step response (d.f.: 222; $t = 2.21$, $p = 0.023$; GLMM; Fig. 6c, inset). This increase in $R_I$ in *Fmr1*−/y mice was associated with an increase in action potential (AP) discharge (Interaction: d.f.: 5, 230; $F = 6.17$; $p < 0.0002$; two-way RM ANOVA, Fig. 6d), resulting from a decreased rheobase currents in the recorded L4 SCs (d.f.: 222; $t = 2.15$, $p = 0.035$; GLMM, Fig. 6d, inset). The dynamic response of neurons to modulating current when measured with a sinusoidal wave of current injection (0.2–20 Hz, 50 pA, 20 s duration, Fig. 6e) led to a resonant frequency of 1.1 ± 0.1 Hz in L4 SCs from *Fmr1*−/y mice, which was higher than that of 0.8 ± 0.1 Hz in WT littermates (d.f.: 25; $t = 3.25$; $p = 0.002$; LMM; Fig. 6f). Furthermore, there was no change in resonant dampening (Q-factor: WT: 1.23 ± 0.07; *Fmr1*−/y; 1.13 ± 0.03; d.f.: 24; $t = 0.7$; $p = 0.49$; T-test) indicating equally sustained activity at these frequencies between genotypes. Further analysis of

passive membrane properties (Supplementary Fig. 6B and C) did not reveal genotype-specific differences. While AP amplitude was minimally reduced (Supplementary Fig. 6E), no other parameter was significantly altered, confirming the specificity of $R_I$ leading to altered cellular excitability. These analyses demonstrate that L4 SCs from *Fmr1*−/y mice are intrinsically more excitable than their WT counterparts.

In S1 L5 pyramidal cells, HCN channel density is reduced leading to reduced $I_h$ as measured indirectly as a voltage sag in current-clamp[17,22]. Therefore, we next asked whether $I_h$ mediated sag is also reduced in L4 SCs and contributes to the genotypic differences in intrinsic excitability we have observed. We first measured the sag and membrane rebound in response to hyperpolarising current steps in current-clamp from −60 mV (0 to −125 pA, 25 pA steps, 500 ms duration; Fig. 7a). The voltage sag, as measured as a percentage of the maximum hyperpolarisation (Fig. 7b) was significantly reduced in *Fmr1*−/y mice (7.6 ± 0.6% of maximum) when compared with WT controls (10.9 ± 0.5% of maximum, d.f.: 218; $t = 3.59$, $p = 0.0003$; GLMM), indicating reduced $I_h$. A further measure of $I_h$ is the rebound

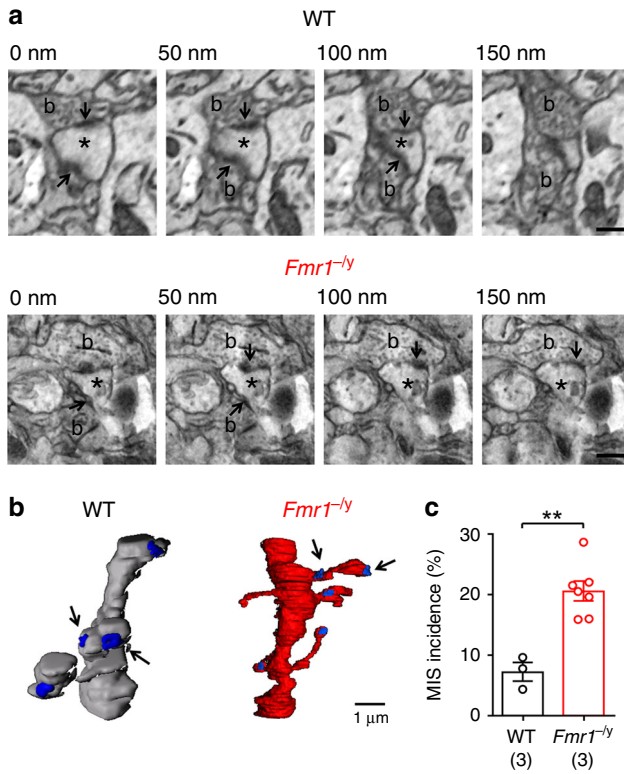

**Fig. 4** L4 spines in $Fmr1^{-/y}$ mice form multiple synaptic contacts. **a** Serial electron micrographs in L4 from WT and $Fmr1^{-/y}$ mice, indicating spines (asterisk) contacted by multiple presynaptic boutons (b) each with a PSD (arrows); scale bar: 500 nm. **b** Reconstructed dendrites from WT (grey) and $Fmr1^{-/y}$ (red) mice, with PSDs (blue) and MIS indicated (arrows). **c** Incidence of MIS in WT and $Fmr1^{-/y}$ mice. Statistics shown: $**p < 0.01$ from unpaired $t$-test. All data are shown as mean ± SEM and source data for all plots are provided as a Source Data file

potential produced on return to −60 mV[22,29]. Consistent with reduced sag, we observed a lower rebound potential in $Fmr1^{-/y}$ L4 SCs when measured relative to the steady-state potential (Fig. 7c). Furthermore, the rebound slope from individual cells was −0.09 ± 0.01 mV/mV in $Fmr1^{-/y}$ neurons, lower than that of WT (−0.11 ± 0.01 mV/mV, d.f.: 207; $t = 2.28$, $p = 0.024$; LMM, Fig. 7d).

We next applied the $I_h$ blocker ZD-7288 (ZD; 20 μM) to a subset of cells to assess the effect of $I_h$ on intrinsic excitability. We observed a tendency to greater $R_I$ in $Fmr1^{-/y}$ than in WT mice (d.f.: 57; $t = 1.85$, $p = 0.078$; LMM; Fig. 7e), similar to that we had observed previously (Fig. 6c). Following ZD application in WT L4 SCs, $R_I$ increased by 49% (d.f.: 28; $t = 6.05$, $p = 1.99 \times 10^{-7}$; LMM; Fig. 7e), while $Fmr1^{-/y}$ L4 SCs only showed a 14% increase (d.f.: 28; $t = 1.28$, $p = 0.20$; LMM; Fig. 7e). The ZD effect on $R_I$ was significantly lower $Fmr1^{-/y}$ L4 SCs compared with WT (d.f.: 57; $t = 4.37$, $p = 6.3 \times 10^{-5}$; LMM; Fig. 7f). Given the observed differences in AP discharge between genotypes (Fig. 6d), we next tested whether ZD normalised this genotypic difference. In WT L4 SCs, ZD application significantly increased AP firing (d.f.: 5, 80; $F = 3.2$; $p = 0.011$ for interaction; two-way RM ANOVA; Fig. 7g). However, ZD had no effect on the AP discharge of $Fmr1^{-/y}$ L4 SCs (d.f.: 5, 174; $F = 0.23$; $p = 0.95$ for interaction; two-way ANOVA; Fig. 7h), consistent with reduced sag. Finally, we examined the effect ZD had on the resonance of L4 SCs. In WT L4 SCs, ZD increased the impedance at low frequencies by 33% (d.f.: 15; $t = 2.66$, $p = 0.017$; GLMM; Fig. 7i, k), whereas ZD had no effect on impedance in $Fmr1^{-/y}$ neurons (d.f.: 13; $t = 0.83$,

$p = 0.41$; GLMM; Fig. 7j, k). These data show that the intrinsic excitability of L4 SCs is increased in $Fmr1^{-/y}$ mice, with WT L4 SC excitability increased by ZD application, potentially explaining genotype-specific differences in cellular intrinsic excitability.

Voltage sag and rebound are indicative of altered $I_h$. To directly measure $I_h$ in L4 SCs we next performed dedicated voltage-clamp experiments using a paradigm described previously[30]. $I_h$ was recorded from L4 SCs in the presence of sodium channel, potassium channel, calcium channel, and GABA$_A$ receptor blockers, as well as AMPA and NMDA antagonists, from −50 mV with hyperpolarising steps (10 mV steps, 5 second duration, Fig. 8a). $I_h$ had a half-maximal activation potential ($V_{1/2\,max}$) in WT L4 SCs of −86 mV, which in $Fmr1^{-/y}$ was more hyperpolarised at −92 mV (d.f.: 4, 584; $F = 4.58$, $p = 0.001$; F-test; Fig. 8b). Despite this difference, $I_h$ elicited at the most hyperpolarised voltage steps was similar (d.f.: 1, 370; $F = 0.001$, $p = 0.97$; F-test), suggesting a normal complement of HCN channels (these currents in both WT and $Fmr1^{-/y}$ L4 SCs were sensitive to ZD, Fig. 8b, inset). As the activation of $I_h$ is directly associated to the intracellular cyclic-AMP concentration[31], we next asked if increasing intracellular cyclic-AMP could rescue $I_h$ activation in $Fmr1^{-/y}$ neurons. To increase cyclic-AMP levels, we bath applied the adenylyl cyclase activator forskolin (50 μM) to the bath. Forskolin significantly increased the activation of $I_h$ in both WT and $Fmr1^{-/y}$ L4 SCs (Fig. 8c), normalising the $I_h$ activation curves between genotypes (d.f.: 4, 310; $F = 0.2$, $p = 0.94$; F-test, Fig. 8d). This data indicates that the decrease in $I_h$ and hence increase in intrinsic excitability, in $Fmr1^{-/y}$ L4 SCs results from a reduced cAMP-mediated shift in HCN activation.

**Enhanced dendritic summation in L4 SCs from $Fmr1^{-/y}$ mice**. Given that NMDARs and HCN channels are a key determinants of dendritic integration[19,20], we next assessed both spatial and temporal dendritic summation in the $Fmr1^{-/y}$ L4 SCs. To address spatial summation in L4 SC dendrites we performed near-simultaneous glutamate uncaging at multiple spines (Fig. 9a), by focal puff application of Rubi-Glu (10 mM) and rapidly uncaged on dendritic spines (0.5 ms/spine). We first performed a sequential uncaging (i.e. each spine individually), then near-simultaneous uncaging of spine ensembles (i.e. groups of spines; Fig. 9b).

Summating EPSPs ultimately resulted in a AP discharge from L4 SCs. $Fmr1^{-/y}$ L4 SCs required activation of fewer spines on average to initiate an AP (d.f.: 23; $t = 2.3$; $p = 0.03$, T-test; Fig. 9c), which was more pronounced when silent spines excluded from analysis (d.f.: 18; $t = 3.2$; $p = 0.005$). In five $Fmr1^{-/y}$ L4 SCs, uncaging at spines individual was not performed, thus were not included in further analysis. Measurement of the summated EPSP, with respect to number of spines near-simultaneously uncaged showed that both WT and $Fmr1^{-/y}$ L4 SC dendrites showed an increase in EPSP amplitude with increasing number of spines (Fig. 9d), which was significantly greater in the $Fmr1^{-/y}$ L4 SCs (d.f.: 1, 170; $F = 8.98$; $p = 0.003$; F-test). This measure will include effects due to increased spine synaptic strength and input resistance, in addition to dendritic integrative properties. Therefore, we next compared the expected linear sum of single-spine EPSPs with that of the observed summated EPSP (Fig. 9e), thereby excluding individual spine strength and input resistance effects on EPSP amplitude. We observed sublinear integration in WT and $Fmr1^{-/y}$ L4 SCs, however, WT neurons showed low levels of integration (Slope: 0.50 ± 0.09), while $Fmr1^{-/y}$ neurons presented over 50% higher summation (Slope: 0.79 ± 0.08; d.f.: 1, 195; $F = 3.18$; $p = 0.044$; F-test). These data clearly show that the dendrites of $Fmr1^{-/y}$ L4 SCs undergo excessive dendritic summation of synaptic inputs. To confirm that dendritic

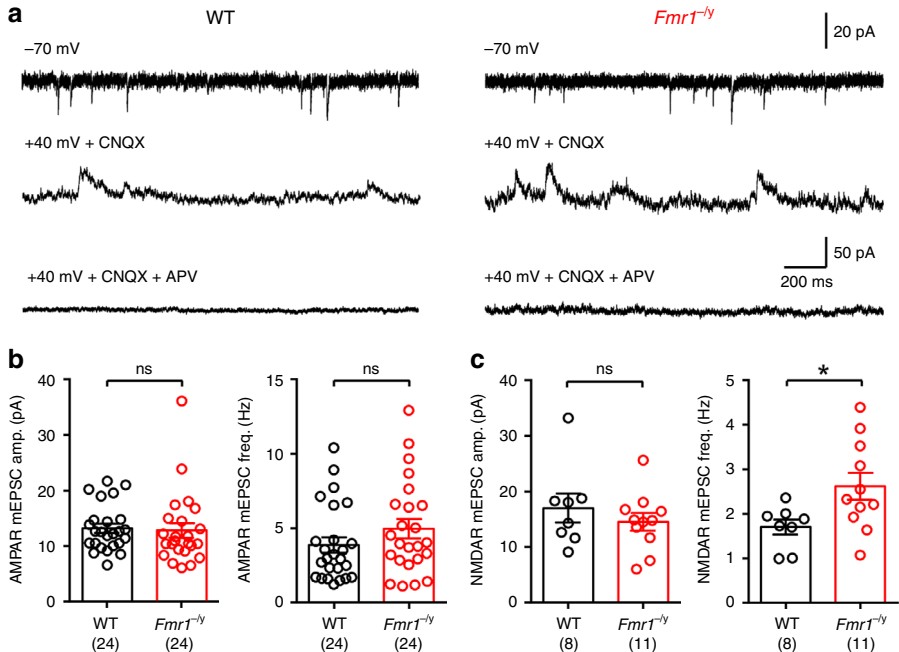

**Fig. 5** mEPSCs in $Fmr1^{-/y}$ L4 SCs show enrichment of NMDAR synapses. **a** mEPSCs recorded from L4 SCs for AMPAR at −70 mV (top), NMDAR at + 40 mV with CNQX (10 µM, middle), and following application of the NMDAR antagonist D-AP5 (50 µM, bottom) in the same cell; from WT (left) and $Fmr1^{-/y}$ (right) mice. **b** Quantification of AMPAR mEPSC amplitude (WT: 13.1 ± 0.8 pA; $Fmr1^{-/y}$; 12.7 ± 1.3 pA) and frequency (WT: 3.9 ± 0.5 Hz; $Fmr1^{-/y}$; 4.9 ± 0.6 Hz) in WT (black) and $Fmr1^{-/y}$ (red) mice. Number of mice indicated in parenthesis. **c** NMDAR mEPSC amplitude (WT: 16.9 ± 2.6 pA; $Fmr1^{-/y}$; 14.4 ± 1.6 pA) and frequency (WT: 1.7 ± 0.17 Hz; $Fmr1^{-/y}$; 2.6 ± 0.3) measured in WT and $Fmr1^{-/y}$ mice. Statistics shown: ns $p > 0.05$, *$p < 0.05$ from unpaired $t$-test. All data are shown as mean ± SEM and source data for all plots are provided as a Source Data file

summation is altered in response to endogenous synaptic transmission, we next provided extracellular stimulation to thalamocortical afferents (TCA) from the ventrobasal thalamus, whilst recording from L4 SCs (Fig. 9f). Stimulus intensity was titrated so that an EPSC of ~150 pA was produced, then trains of EPSPs were elicited in current-clamp at either 5 or 10 Hz. At these stimulation intensities summating EPSPs in L4 SCs in WT mice never produced a somatic AP, however, in $Fmr1^{-/y}$ mice 5 Hz stimulation resulted in an AP in 19 ± 7% of recordings (d.f.: 16; $t = 2.57$ & 3.81; $p = 0.02$ & 0.002, T-test) and 10 Hz stimulation 55 ± 13% of the time (d.f.: 16; $t = 3.81$; $p = 0.002$, T-test), confirming that dendritic integration properties alter the output of L4 SCs, to promote hyperexcitability (Fig. 9g).

As $I_h$ has known effects on dendritic summation[19], we next asked whether ZD altered summation properties. First, we determined whether inhibition of HCN channels altered amplitude or kinetics of synaptic events. Application of ZD itself had no effect on spontaneous EPSC amplitudes, frequencies, or kinetics (Supplementary Fig. 8). However, spontaneous EPSCs were of higher frequency in $Fmr1^{-/y}$ L4 SCs, potentially indicating underlying circuit hyperexcitability (d.f.: 25; $t = 2.99$, $p = 0.016$; GLMM). Summating uEPSPs from WT mice (normalised to the initial uEPSP) displayed long decay times at low summation, which were more rapid at higher summation levels (Supplementary Fig. 9A, B). By comparison, in $Fmr1^{-/y}$ mice we did not observe this relationship and the genotype-specific log (EPSP summation) was divergent (d.f.: 1, 109; $F = 32.1$, $p < 0.0001$; F-test). The summation-dependent temporal sharpening of EPSPs in WT neurons was abolished following application of ZD (Comparing slope: d.f.: 1, 85; $F = 6.4$, $p = 0.01$; F-test; Supplementary Fig. 6D) and also prolonged decay times of the first EPSP (Fig. 9f, d.f.: 15; $t = 2.34$; $p = 0.034$; T-test; Supplementary Fig. 9C). ZD had no observable effect on summating EPSPs in $Fmr1^{-/y}$ L4 SCs (Supplementary Fig. 9E). Finally to

confirm that altered $I_h$ and NMDAR function contribute to the observed aberrant dendritic summation, in a subset of experiments we examined the effects of both ZD and AP-5 on EPSP summation during multispine uncaging. Application of either ZD or AP-5 to near-simultaneous uncaging of uEPSPs in WT L4 SCs had minimal effect on the observed summation when compared with the expected linear sum (Supplementary Fig. 10A), consistent with an absence of non-linear summation. However, bath application of either ZD or AP-5 significantly reduced the summation of $Fmr1^{-/y}$ L4 SCs (Supplementary Fig. 10B). These findings confirm that both reduced HCN activation and increased NMDARs contribute to the enhanced summation in dendrites of $Fmr1^{-/y}$ L4 SCs relative to WT cells.

## Discussion

L4 of the primary somatosensory cortex is the first layer to receive and integrate incoming sensory information, which is integrated and relayed within the cortex. As such, L4 SCs play a crucial role in sensory perception[14]. Individuals with FXS show altered sensory processing[32,33] and mouse models show altered circuit processing in primary sensory areas[1,15,17,18,34,35]. Furthermore, while FMRP has been shown repeatedly to regulate synapse function and plasticity, little is known about how these alterations affect dendritic spine function and dendritic integration to sensory input. To address these questions, we used glutamate uncaging at L4 SC dendritic spines to examine how they integrate and generate action potentials following synaptic stimulation. We show that L4 SCs in S1 have dendritic and synaptic properties that result in increased action potential generation in $Fmr1^{-/y}$ mice relative to WT controls. Specifically, we show increased excitatory synaptic currents at individual spines resulting from increased AMPAR and NMDAR content. Despite this, we observed no change in spine morphology using STED microscopy

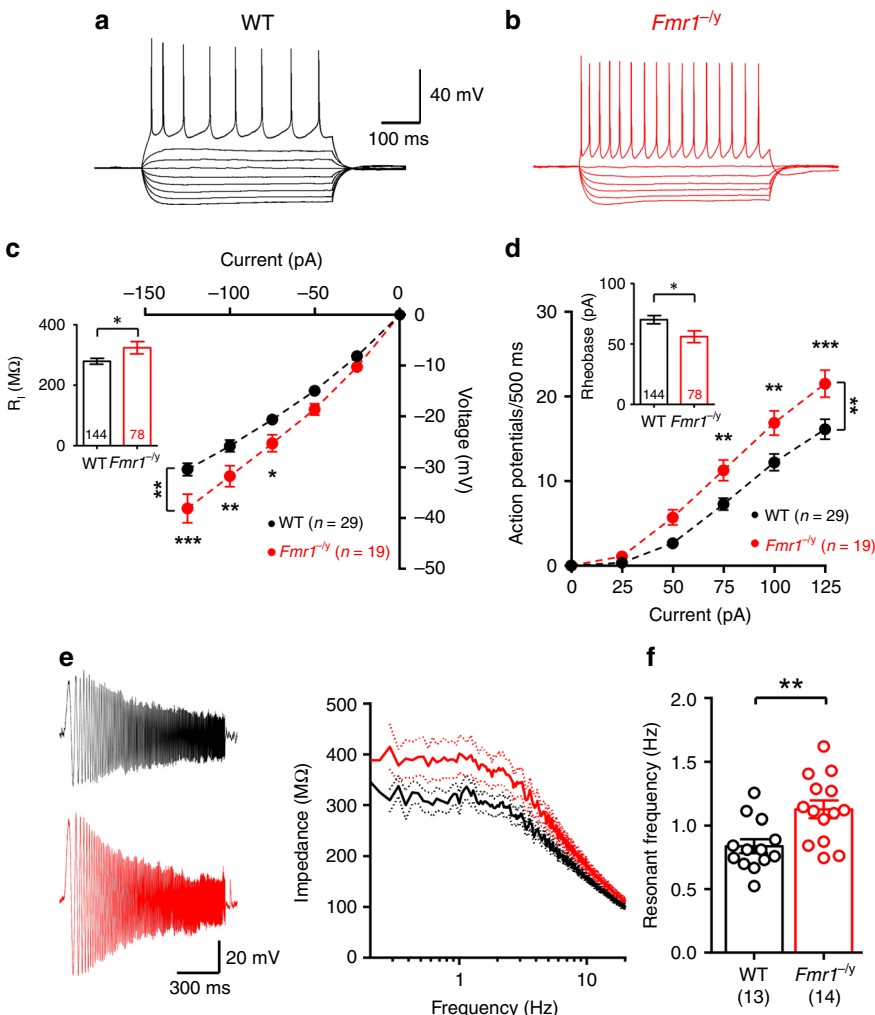

**Fig. 6** Altered intrinsic physiology of L4 SCs in *Fmr1*$^{-/y}$ mice. Voltage responses to hyper- and depolarizing current steps (−125 to +125 pA, 25 pA steps, 500 ms duration) led to AP discharge in WT (**a**) and *Fmr1*$^{-/y}$ (**b**) mice. **c** The current–voltage response to hyperpolarizing currents with linear fit (dashed lines) in WT (black) and *Fmr1*$^{-/y}$ (red) mice. **c** (inset) $R_I$ measured from all L4 SCs tested. **d** Current–frequency plot showing AP discharge. **d** (inset) Average rheobase current measured in all cells. **e** Subthreshold membrane chirps (0.2–20 Hz, 50 pA, 20 s duration) in L4 SCs from WT (black) and *Fmr1*$^{-/y}$ mice. Right, frequency–impedance plot for both genotypes ± SEM, shown on a logarithmic frequency scale. **f** Resonant frequency of L4 SCs from both genotypes. Statistics shown: \*$p < 0.05$, \*\*$p < 0.01$, \*\*\*$p ± < 0.001$, from LMM (**c** and **d** insets, **f**) and two-way ANOVA (**c** and **d**, main). Summary plots of all cells recorded for **c** (inset) and **d** (inset) can be found in Supplementary Fig. 5. All data are shown as mean ± SEM and source data for all plots are provided as a Source Data file

and there was little correlation between spine structure and function, indicating that spine morphology is not an effective proxy for spine function, at least at the age used in this study. However, electron microscopic analysis revealed an increase in multiply innervated spines which likely accounts for the increase in single-spine synaptic currents. Interestingly, there was also an increase in silent spines which agrees with the increase in NMDAR mEPSC frequency, but not AMPAR mEPSC frequency. The overall increase in dendritic spine currents was accompanied by enhanced dendritic integration likely resulting, at least in part, from a ~50% reduction in $I_h$. This reduced $I_h$ was causal to the altered intrinsic physiology of L4 SCs at P12–14. Finally, TCA stimulation at frequencies that fail to elicit AP discharge from L4 SCs in WT mice, in the presence of intact synaptic inhibition, reliably elicits APs in *Fmr1*$^{-/y}$ neurons, indicating that the local inhibitory circuit cannot compensate for the increase in synaptic and dendritic excitability. Together these findings demonstrate that aberrant dendritic spine function and dendritic integration combine to result in cellular hyperexcitability in L4 SCs. As the first cortical cells to receive input from the sensory periphery, the

resultant hyperexcitability likely contributes previously reported circuit excitability in *Fmr1*$^{-/y}$ mice and the sensory hypersensitivities in individuals with FXS.

Our study quantifies the incidence of MIS in intact tissue and implicates their presence in pathological states associated with disease models. Indeed, the mean increase in spine uEPSC amplitude, but not miniature, spontaneous or unitary EPSCs, in *Fmr1*$^{-/y}$ mice is likely caused by the increase in the number of MIS. Indeed, the presence of MIS in both WT and *Fmr1*$^{-/y}$ mice disagrees with the one spine/one synapse hypothesis[36]. A potential mechanistic link between loss of FMRP and the increase in MIS may come from its ability to regulate PSD-95. *Psd-95* mRNA is a known FMRP target[37] and an increase in PSD-95 puncta in L4 of S1 has been observed[7] with no change in cell number, dendritic morphology, or spine density in *Fmr1*$^{-/y}$ mice[26]. Furthermore, transient overexpression of PSD-95 results in increased MIS incidence through nitric oxide synthase, as well as NMDARs and other LTP mechanisms[22,25,38–40]. Future experiments exploring the effect of NOS blockade, PSD-95, and NMDAR function in *Fmr1*$^{-/y}$ mice should test the mechanism of

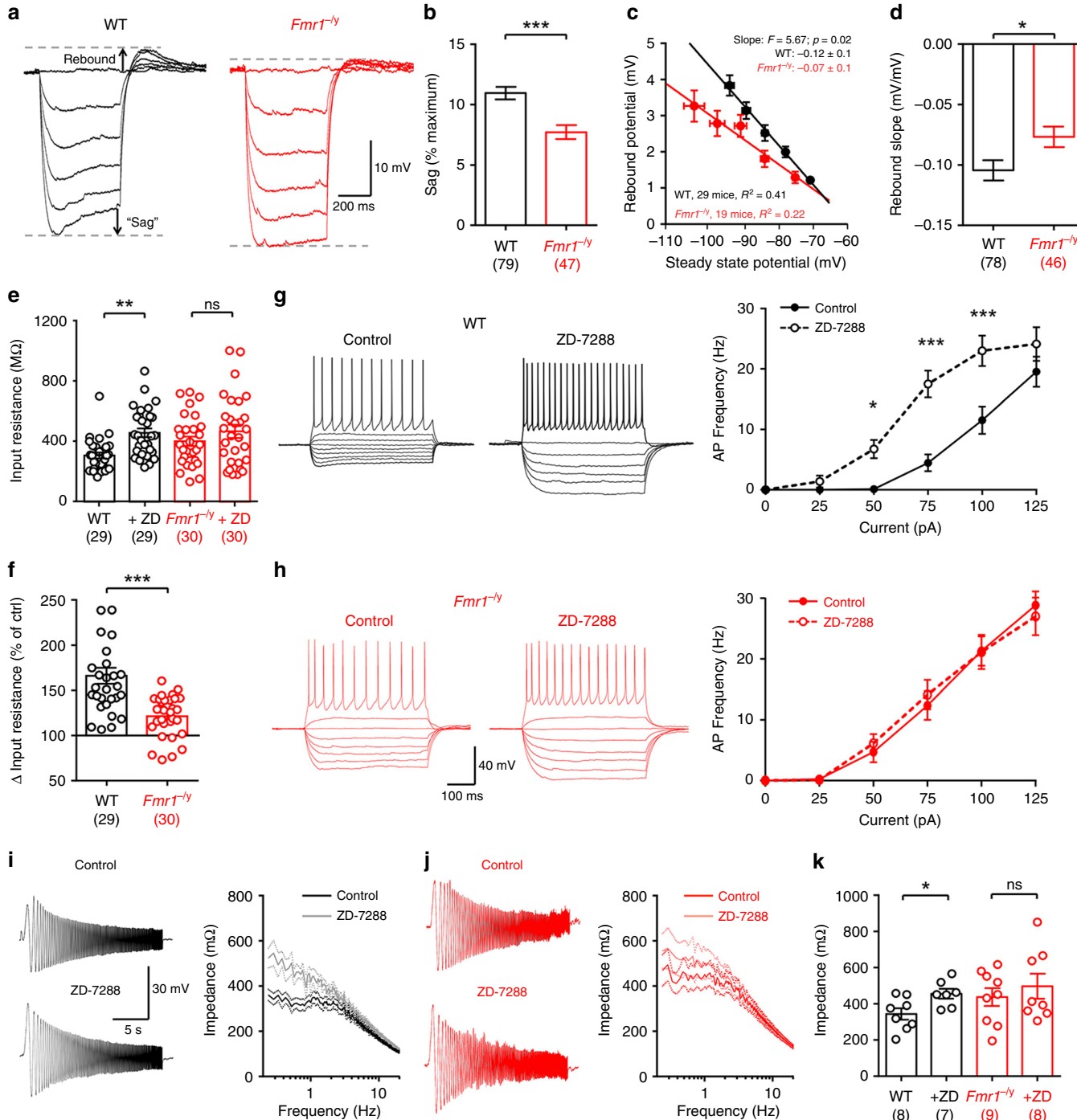

**Fig. 7** $I_h$ is reduced in L4 SCs from $Fmr1^{-/y}$ mice, resulting in hyperexcitability. **a** Hyperpolarizing steps in L4 SCs (0 to −125 pA, 25 pA steps, 500 ms duration) with voltage "sag" and rebound potential indicated, as measured in WT (black, left) and $Fmr1^{-/y}$ mice (red, right). **b** Quantification of voltage sag expressed as % of maximum voltage for WT and $Fmr1^{-/y}$ L4 SCs **c** plot of rebound potential, as a function of steady-state voltage for WT and $Fmr1^{-/y}$S L4 SCs, fitted with linear regression and with fit values displayed. **d** quantification of the rebound slope of individual L4 SCs for both genotypes. **e** $R_I$ measured before and after bath application of the $I_h$ blocker ZD-7288 (ZD; 20 μM) in WT and $Fmr1^{-/y}$ L4 SCs. **f** Change in $R_I$ change following ZD application (as 100% of control levels). **g** (left) Hyper- to depolarising current steps (−125 to +125 pA, 25 pA steps, 500 ms duration) in WT L4 SCs before and after ZD application. **g** (right) Current–frequency plot of AP discharge before (solid lines) and after (dashed lines) ZD application. **h** The same analysis as in (**g**), but in $Fmr1^{-/y}$ L4 SCs. **i** Subthreshold membrane chirps (0.2–20 Hz, 50 pA, 20 s duration) and current–impedance plot for WT L4 SCs before (black) and after (grey) ZD application. **j** The same data as in (**f**), but in $Fmr1^{-/y}$ mice. **k** Impedance measured at peak resonant frequency in WT and $Fmr1^{-/y}$ L4 SCs before and after ZD (+ZD) application. Statistics shown: ns $p > 0.05$, *$p < 0.05$, **$p < 0.01$, ***$p < 0.001$, from LMM (**b**, **d**, **e**, **f**, **k**). Summary plots of all data shown in (**b**) and (**d**) can be found in Supplementary Fig. 7. All data are shown as mean ± SEM and source data for all plots are provided as a Source Data file

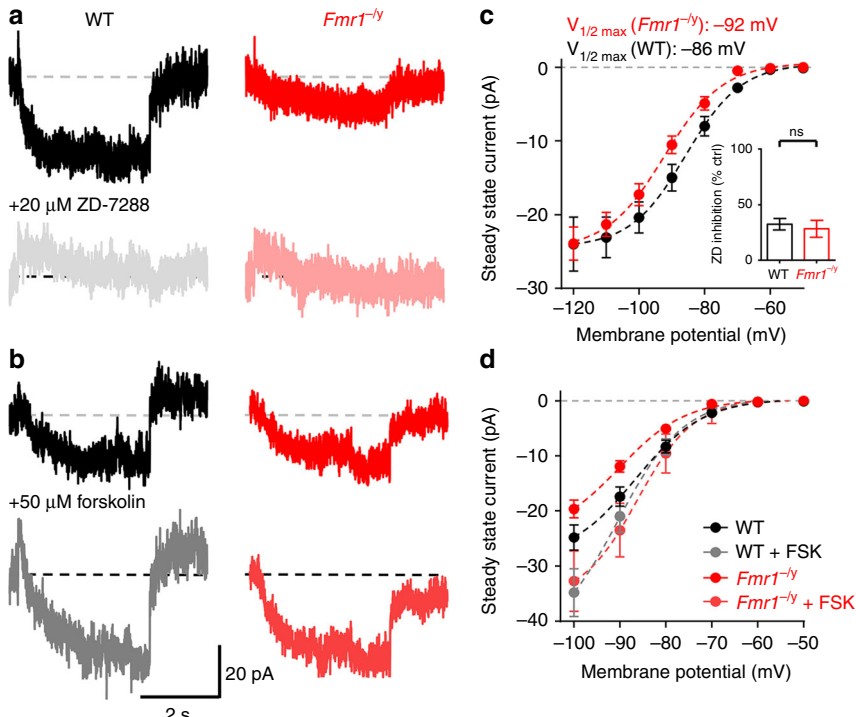

**Fig. 8** Altered $I_h$ voltage sensitivity in $Fmr1^{-/y}$ L4 SCs, due to reduced cyclic-AMP. **a** Subtracted $I_h$ traces recorded during a −50 mV step from −50 mV holding potential for WT (black) and $Fmr1^{-/y}$ (red) L4 SCs, and following ZD application (grey, light red, respectively). **b** $I_h$ measured over the range of −50 to −120 mV for both WT and $Fmr1^{-/y}$ L4 SCs fitted with a sigmoidal curve (dashed lines). $V_{1/2\,max}$ is indicated. Inset, $I_h$ was blocked to a similar degree by ZD in both genotypes when tested on steps to −100 mV. **c** $I_h$ recorded before (top) and after (bottom) application of forskolin. **d** Quantification of $I_h$ responses over the range of −50 to −100 mV, fitted with a sigmoidal curve. All data are shown as mean ± SEM and source data for all plots are provided as a Source Data file

MIS formation and influence on dendritic protein synthesis, as well as potential therapeutic targeting.

Interestingly, the increase in spines with increased uEPSC amplitudes and MIS was mirrored by an increase in silent spines, though their number was insufficient to compensate for the overall increase in dendritic currents in other spines. An increase in silent TCA synapses at P7[18] was previously reported in $Fmr1^{-/y}$ mice. However, this study also reported a delay in the critical period for inducing LTP at these synapses which terminated at P10. Therefore, the period of synaptic potentiation at TCA synapses is complete by the age we tested in this study. Hence the percentage of silent spines receiving TCA input would be expected to be low[41]. Furthermore, the reduced connectivity between L4 SCs at P12–14, despite no change in spine density[26], strongly indicates that SC to SC synapses are preferentially silent at this developmental stage in the $Fmr1^{-/y}$ mouse. Together, these findings suggest that silent spines measured in our study reflect cortico-cortical, rather than TCA, synapses. Given the hierarchical nature of sensory system development, it would not be surprising if a delay in intra-cortical synapse development in $Fmr1^{-/y}$ mice follows the aforementioned delay in TCA synapse development, but this remains to be directly tested.

While dendritic spines are functionally disrupted in the $Fmr1^{-/y}$ mouse, using super-resolution microscopy we found no evidence of a genotypic difference in spine morphology of L4 SC neurons. This is in good agreement with our previous findings that spine morphology is unaffected in hippocampal CA1 and layer 5 S1 neurons[6]. Furthermore, we find only a weak correlation between dendritic spine structure and function, demonstrating the pitfalls of using spine structure as a proxy for synaptic function, especially in young animals and genetic models of disease. These findings are in stark contrast to those observed

from post-mortem human tissue[3] or from other mouse studies[5]; however, these studies were only performed with diffraction-limited microscopy, suggesting that super-resolution imaging techniques should be the gold-standard for dendritic spine morphological studies in future. Single dendritic spines do not typically produce AP discharge from neurons, rather they require co-activation and summation of multiple synaptic inputs arriving with high temporal precision[42]. L4 SCs have been previously been shown to possess linear integration of $Ca^{2+}$ influx in their dendrites[43]. We show that synaptic potentials sublinearly integrate in L4 SCs of WT mice, and that this integration is strongly enhanced in $Fmr1^{-/y}$ mice, leading to more efficient discharge of APs, due in large part to a combination of increased NMDARs and reduced $I_h$. The latter has been implicated in the altered neuronal excitability of FXS[17,22], with the HCN1 channel expression dictating whether the current is increased or decreased. Unlike these former studies, we provide evidence that $I_h$ is not reduced in L4 SCs, but rather displays shifted activation properties, likely due to reduced cyclic-AMP levels. This finding in in agreement with previous work implicating altered cAMP levels in the aetiology of FXS[44–48]. Whether the altered $I_h$ currents in the absence of FMRP reported in other cell types[17,22] could also be explained by altered cAMP levels is not known; however, at least for layer 5 neurons in somatosensory cortex, a reduced level of HCN channels has also been reported[17]. Future experiments will be needed to determine the developmental and cell-specific nature of cellular hyper-excitability in $Fmr1^{-/y}$ mice.

Our observations showing sublinear dendritic integration in layer 4 SCs are at odds with reported NMDAR-dependent non-linear (supra-linear) summation of cortical cells reported from many laboratories[20,21,49,50]. However, many factors may account for this discrepancy, including recording conditions, stimulation

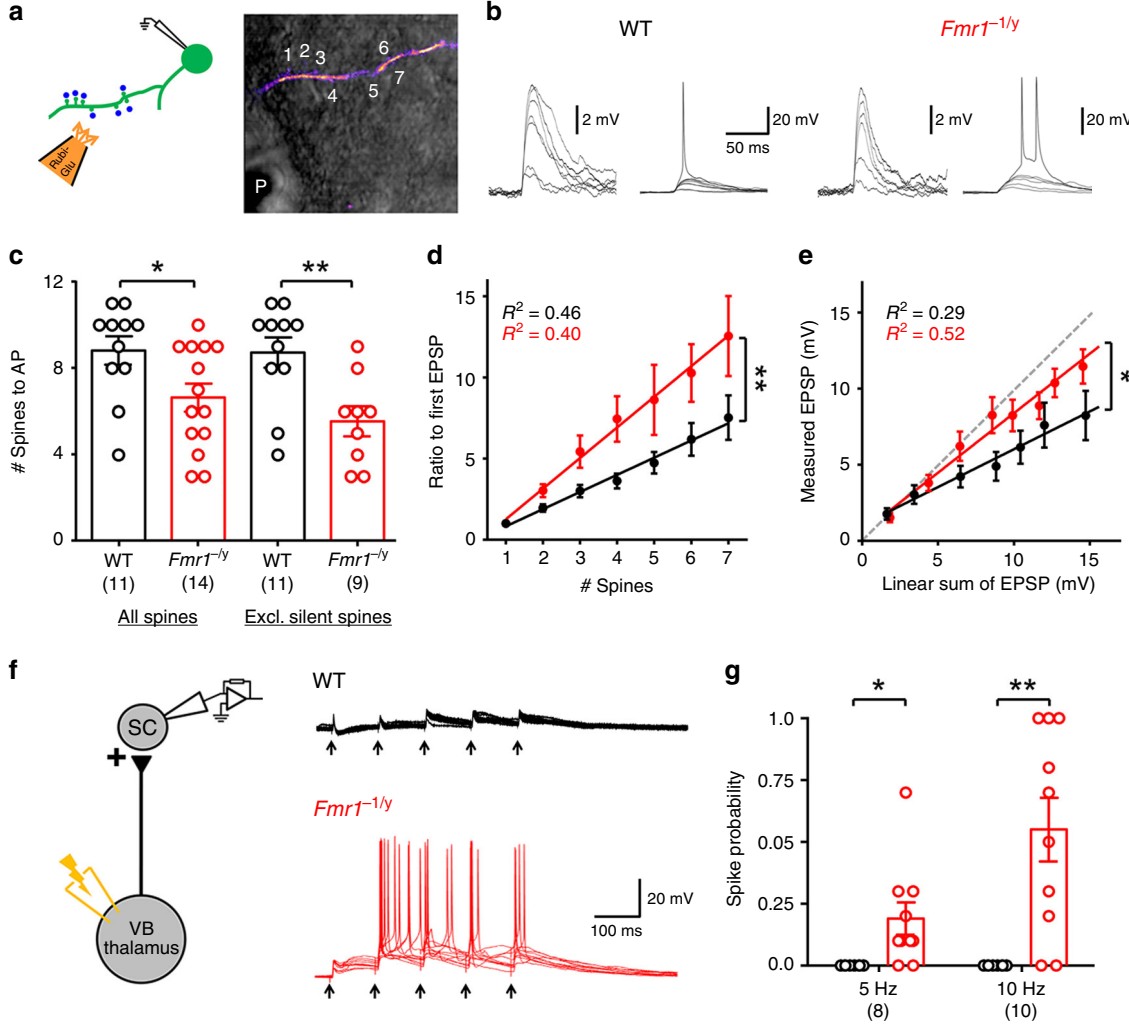

**Fig. 9** Enhanced dendritic integration of L4 SCs in *Fmr1*⁻/ʸ mice. **a** Schema of near-simultaneous glutamate uncaging (Rubi-Glu) at multiple spines (blue dots/numbers). **b** Near-simultaneous glutamate uncaging produced subthreshold (inset, right) and suprathreshold uEPSPs (inset, left) along dendrites. **c** The number of spines required to evoke an AP, from all spines (left; WT: 8.8 ± 0.7; *Fmr1*⁻/ʸ; 6.6 ± 0.6) and excluding "silent spines" (right; WT: 8.7 ± 0.7; *Fmr1*⁻/ʸ; 5.6 ± 0.7). **d** Summation of near-simultaneous subthreshold uEPSPs normalized to the first EPSP in WT (black) and *Fmr1*⁻/ʸ (red) L4 SCs (Slope: WT: 1.1 ± 0.13; *Fmr1*⁻/ʸ; 1.9 ± 0.2; d.f.: 1, 170; $F = 8.98$; $p = 0.003$; F-test). **e** Summating uEPSPs plotted against the expected linear sum. Unity is indicated (grey). **f** Electrical stimulation of TCA at low frequency 10 Hz is shown. **g** Average spike probability in response to 5 and 10 Hz stimulation. Statistics shown: *$p < 0.05$, **$p < 0.01$. All data are shown as mean ± SEM and source data for all plots are provided as a Source Data file

paradigms, cell type and developmental age. Furthermore, the somatosensory cortex has a well-described developmental profile of membrane properties, notably decreasing membrane resistance as a function of age[51]. This combined with the compact dendritic arbour of L4 SCs[26], will lead to these neurons at the age of ~14 days likely having very uniform cable properties[28]. It is possible that as L4 SCs mature, their dendrites may develop non-linear properties. Irrespective of the differences between studies, we provide the first direct evidence in *Fmr1*⁻/ʸ neurons for a functional deficit at excitatory synapses onto dendritic spines and that these alterations contribute to an increase in dendritic integration. The summation of synaptic responses contributes to hyperexcitability of sensory neurons in the *Fmr1*⁻/ʸ mouse, which along with changes in intrinsic excitability, may underlie pathophysiology associated with altered sensory function.

## Methods

**Animals and ethics**. All procedures were performed in line with Home Office (ASPA, 2013; HO license: P1351480E) and institutional guidelines. All experiments were performed on C57/Bl6J mice, bred from *Fmr1*⁺/⁻ mothers, cross-bred with *Fmr1*⁺/ʸ male mice, giving a Mendelian 1:1 ratio of *Fmr1*⁺/ʸ and *Fmr1*⁻/ʸ amongst male offspring. Only male mice were used for the present study and all mice were killed at P10–15, before separation from the mother. Mothers were given ad libitum access to food and water and housed on a 12 h light/dark cycle. All experiments and analysis were performed blind to genotype.

**Acute slice preparation**. Acute brain slices were prepared similar to previously described[52,53]. Briefly, mice were decapitated without anaesthesia and the brain rapidly removed and placed in ice-cold carbogenated (95% $O_2$/5% $CO_2$) sucrose-modified artificial cerebrospinal fluid (in mM: 87 NaCl, 2.5 KCl, 25 NaHCO₃, 1.25 NaH₂PO₄, 25 glucose, 75 sucrose, 7 MgCl₂, 0.5 CaCl₂). 400 μm thick thalamo-cortical (TC) slices were then cut on a Vibratome (VT1200s, Leica, Germany) and then stored submerged in sucrose-ACSF warmed to 34 °C for 30 minutes and transferred to room temperature until needed.

**Whole-cell patch-clamp recordings**. For electrophysiological recordings slices were transferred to a submerged recording chamber perfused with carbogenated normal ACSF (in mM: 125 NaCl, 2.5 KCl, 25 NaHCO₃, 1.25 NaH₂PO₄, 25 glucose, 1 MgCl₂, 2 CaCl₂) maintained at near physiological temperatures (32 ± 1 °C) with an inline heater (LinLab, Scientifica, UK) at a flow rate of 6–8 ml/min. Slices were visualized with IR-DIC illumination (BX-51, Olympus, Hamburg, Germany) initially with a ×4 objective lens (N.A. 0.1) to position above a L4 barrel, and then with a ×20 water-immersion objective (N.A. 1.0, Olympus). Whole-cell patch-clamp recordings were made with a Multiclamp 700B amplifier (Molecular Devices, USA). Recording pipettes were pulled from borosilicate glass capillaries

(1.7 mm outer/1 mm inner diameter, Harvard Apparatus, UK) on a horizontal electrode puller (P-97, Sutter Instruments, CA, USA), which when filled with intracellular solution gave a pipette resistance of 4–5 MΩ. Unless otherwise stated, all V-clamp recordings were performed at $V_M = -70$ mV. All signals were filtered at 10 kHz using the built in 4-pole Bessel filter of the amplifier, digitized at 20 kHz on an analogue-digital interface (Digidata 1440, Axon Instruments, CA, USA), and acquired with pClamp software (pClamp 10, Axon Instruments, CA, USA). Data were analysed offline using the open source Stimfit software package[54] (http://www.stimfit.org). Cells were rejected if the $I_{hold}$ was > 150 pA in voltage clamp, membrane potential more depolarised than −50 mV in current-clamp, series resistance > 30 MΩ, or the series resistance changed by more than 20% over the course of the recording.

**Sequential dendritic spine 2-photon glutamate uncaging.** Slices were transferred to the recording chamber, which was perfused with normal ACSF, containing 50 µM picrotoxin (PTX) and 300 nM tetrodotoxin (TTX). For voltage-clamp recordings of dendritic spine uncaging neurons were filled with an internal solution containing (in mM): 140 Cs-gluconate, 3 CsCl, 0.5 EGTA, 10 HEPES, 2 Mg-ATP, 2 Na₂-ATP, 0.3 Na₂-GTP, 1 phosphocreatine, 5 QX-314 chloride, 0.1% biotinoylated-lysine (Biocytin, Invitrogen, UK), and 0.1 AlexaFluor 488 or 594 (Invitrogen, UK), corrected to pH 7.4 with CsOH, Osm = 295–305 mOsm. Whole-cell patch-clamp was then achieved and cells allowed to dye fill for 10 min prior to imaging. During this period, we collected 5 min of spontaneous recording, to analyse mEPSCs from recorded neurons at −70 mV voltage clamp. For all imaging and uncaging experiments we used a galvanometric scanning 2-photon microscope (Femto2D-Galvo, Femtonics, Budapest, Hungary) fitted with a femtosecond aligned, tuneable wavelength Ti:Sapphire laser (Chameleon, Coherent, CA, USA), controlled by a Pockel cell (Conoptics, CT, USA). Following dye filling, a short, low zoom z-stack was collected (2 µm steps, 2–3 pixel averaging, 512 × 512 pixels) over the whole dendritic extent of the cell at low laser power (< 5 mW) with a high numerical aperture ×20 lens (N.A. 1.0, Olympus, Japan). Then a short section of spiny dendrite, 50–100 µm from the cell somata, within the top 50 µm of the slice, and running parallel to the slice surface was selected and imaged at higher zoom. Between 7 and 10 spines were then selected based on being morphologically distinct from neighbouring spines, ordered distal to proximal to soma, and then 300 µM Rubi-Glutamate (Rubi-Glu; Ascent Scientific, Bristol, UK) was applied to the bath, and recirculated (total volume: 12.5 ml; flow rate: 6–8 mls/min). Following wash-in of Rubi-Glu (< 2 min), short duration, high-power laser pulses (1 ms, λ780 nm, 80–100 mW, 0.2 µm diameter) local photolysis was performed ~1 µm adjacent to individual spines. In a subset of recordings from WT mice, we confirmed spatial, quantal release, and pharmacological properties of Rubi-Glu uncaging under our recording conditions (Supplementary Fig. 1). Individual spines were sequentially uncaged at 2 s intervals followed by a 40 s pause; therefore each spine receiving Rubi-Glu photolysis every 60 s. All spines underwent photolysis at least three times and the average uncaging-EPSC (uEPSC) at −70 mV measured. In a subset of experiments we confirmed that these uEPSCs were mediated by direct activation of AMPARs by subsequent application of 10 µM CNQX to the perfusing ACSF (Supplementary Fig. 1D). Following each three repetition cycle, the focal plane and dendritic health was checked with short scans, at low power (< 5 mW) to prevent background photolysis. Following successful recording of AMPA uEPSCs, we increased the holding potential to + 40 mV and recorded the outward mixed AMPA/NMDAR currents. In a subset of experiments we confirmed the AMPAR and NMDAR dependence of these outward currents by bath applying 10 µM CNQX and then 50 µM D-AP5 (Supplementary Fig. 1E). AMPA uEPSCs were measured over the first 10 ms following the uncaging stimulus (0.5 ms peak average) at both −70 and + 40 mV. NMDA currents were measured from 20 to 50 ms post-photolysis, which was confirmed to be following complete decay of the AMPA uEPSC at −70 mV. All sequential spine uncaging experiments were performed as quickly as possible following dye filling, to prevent phototoxic damage to the recorded neurons, and L4 SCs resealed with an outside-out patch. Cells were rejected if photolysis resulted in blebbing of dendrites or depolarisation of the membrane potential.

In a subset of experiments, we performed mEPSC analysis of L4 SCs independent of Rubi-Glu photolysis, under the same conditions as above (with no AlexaFluor dye), recording 5 min of mEPSCs at −70 mV voltage clamp. Cells were then depolarised to + 40 mV voltage clamp and mixed AMPA/NMDA mEPSCs recorded for 1 min, after which 10 µM CNQX was applied to the bath. Following full wash-in of CNQX (~2–3 min) a further 5 min of pure NMDA mEPSCs were recorded. In all experiments 50 µM AP-5 was then bath applied, to confirm that the mEPSCs recorded were NMDAR-mediated. All mEPSC data was analysed using a moving-template algorithm[55], with templates made from the tri-exponential non-linear fit to optimal mEPSCs at each holding potential using the event-detection interface of Stimfit. For mEPSCs at −70 mV, the minimum time between EPSCs was set to 7.5 ms, and 25 ms for those at + 40 mV. Detected events were analysed if they had an amplitude greater than 3× the SD of the 5 ms preceding baseline of the mEPSC.

HCN-mediated currents were measured as previously reported[30]. Briefly, slices were transferred to the recording chamber perfused with modified recording ACSF (in mM): 115 NaCl, 5 KCl, 25 NaHCO₃, 1.2 NaH₂PO₄, 2 glucose, 1 MgCl₂, 2 CaCl₂) which was supplemented with channel blockers TEA (5 mM), CdCl₂ (0.1 mM),

BaCl₂ (1 mM), 4-aminopyridine (1 mM), and TTX (300 nM); and blockers for ionotropic receptors CNQX (10 µM), AP-5 (50 µM), and picrotoxin (50 µM), with a flow rate of 4–6 ml/min at room temperature. Cells were recorded with K-gluconate based intracellular solution (in mM: 142 K-gluconate, 4 KCl, 0.5 EGTA, 10 HEPES, 2 MgCl₂, 2 Na₂-ATP, 0.3 Na₂-GTP, 10 phosphocreatine, 0.1% Biocytin, corrected to pH 7.4 with KOH, Osm = 295–305 mOsm). $I_h$ was recorded in voltage clamp from a holding potential of −50 mV and activated by applying hyperpolarising voltage steps (−10 mV, 5 s duration). $I_h$ was measured as the difference in peak to steady-state current during the hyperpolarising step over the full range of potentials. In subsets of experiments, the HCN channel blocker ZD-7288 was bath applied (20 µM) to confirm the identity of the current or the adenylyl cyclase activator forskolin (50 µM) was bath applied. Currents were plotted and fitted with a variable slope sigmoidal function to determine the 50% maximum activation. Representative traces are shown as P/N subtractions of the −10 mV from the −50 mV step.

Summation of thalamic inputs to L4 SCs was measured by electrical stimulation of the ventrobasal thalamus with a twisted bipolar Ni-Chrome wire. Synaptically coupled barrels were identified by placing a field electrode (a patch electrode filled with ACSF) in visually identified barrels and stimulating the thalamus. When a field response was observed, then a L4 SC was recorded in whole-cell patch-clamp with K-gluconate internal solution, as described above. Trains of 5 stimuli were then delivered at 5–10 Hz, with a stimulation intensity sufficient to produce an EPSC of large amplitude similar between genotypes (20–540 pA; WT: 181 ± 35 pA; $Fmr1^{-/y}$: 159 ± 34 pA; d.f.: = 23, $t = 0.44$, $P = 0.66$, T-test). In current-clamp, the EPSP summation was assessed as the ability of the recorded cell to fire an AP in response to this stimulus. Data are shown as the average $P_{spike}$ from 10 trials.

**Near-simultaneous dendritic spine 2-photon glutamate uncaging.** To determine the summation properties of dendrites in L4 SCs we performed near-simultaneous photolysis of Rubi-Glu at multiple dendritic spines[20,49]. Using a current-clamp optimized K-gluconate based internal solution supplemented with 0.1 AlexaFluor 488 (Invitrogen, UK) we dye filled neurons as for sequential photolysis described above, in normal ACSF containing PTX and TTX, but not Rubi-Glu. Once dye filling was complete (<10 min) we imaged the L4 SC (as above) at low zoom, then identified a superficial spiny dendrite 50–100 µm from the soma. At this point we placed a wide puff-pipette (borosilicate patch pipette with tip broken to ~20 µm diameter) just above the surface of the slice, adjacent to the dendrite of interest. The puff-pipette was filled with 10 mM Rubi-Glu in a HEPES buffered ACSF (in mM: 140 NaCl, 2.5 KCl, 10 HEPES, 1.25 NaH₂PO₄, 25 glucose, 1 MgCl₂, 2.5 CaCl₂; adjusted to pH 7.4 with HCl). At this point the dendrite was imaged at high magnification and 7–10 spines chosen and a very low pressure stimulus given to the puff-pipette (3–5 mBar), sufficient to cause dialysis of the Rubi-Glu, but not powerful enough to cause obvious movement of the tissue. The dialysis of Rubi-Glu was maintained throughout the remainder of the recording. The cell was then switched to current-clamp mode, membrane potential held at −60 mV with a bias current, and spines 1–7 sequentially uncaged (0.5 ms laser duration, 80 mW power) to give the individual spines uEPSP amplitude. Following three repetitions and correction of focus, a line scan was created, with 0.5 ms dwell time at each spine ROI in order from distal to proximal. Spines were then uncaged in a cumulative manner, with 1, 2, 3 … n spines uncaged near-simultaneously. The total duration of uncaging was 5.5 ms for 10 spines and there was a 10 s delay between each run of photolysis, with the total protocol lasting minimally 4–5 min. At least three repetitions of this protocol were run and focus re-checked. In a subset of experiments the HCN inhibitor ZD was applied to the perfusing ACSF and a further three repetitions collected. All uEPSP data was analysed as peak amplitude measured over the 20 ms directly following beginning of the photolysis stimuli. Data was either normalised to the first EPSP amplitude, or measured as the absolute simultaneous uEPSP, as plotted against the summed individual uEPSP amplitude for the same spines.

In a set of experiments (without PTX, TTX or AlexaFluor 488), intrinsic electrophysiological properties of L4 SCs were measured, also in current-clamp mode. From resting membrane potential a hyper- to depolarizing family of current injections (−125 to +125 pA, 500 ms duration) were given to the recorded neuron. The input resistance, rheobase current, and action potential discharge frequency were all measured from triplicate repetitions. In a further subset of experiments, 3× series of voltage steps were given (in voltage clamp) from −60 mV to −110 mV (10 mV steps, 500 ms duration) to estimate the amplitude of $I_h$ in the recorded L4 SCs. ZD was then applied to the bath and the same steps repeated. $I_h$ was estimated as the amplitude of the current produced in response to hyperpolarizing voltage steps.

**Visualisation and STED microscopy of recorded neurons.** Following completion of experiments and resealing of the neuron, slices were immediately immersion fixed in 4% paraformaldehyde (PFA) overnight at 4 °C. Slices were then transferred to phosphate buffered saline (PBS; 0.025 M phosphate buffer + 0.9% NaCl; pH: 7.4) and kept at 4 °C until processed (< 3 weeks). Slices were then cryoprotected in a solution containing 30% sucrose in PBS overnight at 4 °C and then freeze-thaw permeablised on lN₂, and returned to cryoprotectant solution for 1–2 h. The slices were then mounted, recording side up, on the stage of a freezing microtome; which had been prepared with a plateau of Optimal Cutting Temperature (OCT) medium

and slices embedded within OCT prior to sectioning. The OCT block containing the recorded slice was trimmed to the slice surface and then 50 μm sections taken from the top 200 μm. The sections were rinsed three times in PBS and then incubated with streptavidin conjugated to AlexaFluor488 (1:500, Invitrogen, UK) at 4 °C for 3–5 days. The slices were then washed for 2 h in repeated washes of PBS, and then desalted with PB and mounted on glass slides with fluorescence protecting mounting medium (Vectorshield, Vector Labs, UK).

Sections were imaged on a gated-stimulated emission-depletion (STED) microscope (SP8 gSTED, Leica, Germany). Cells were found using epifluorescent illumination (488 nm excitation) under direct optics at low magnification (×20 air immersion objective lens, N.A. 0.75) and then positioned under high magnification (×100 oil-immersion objective lens, N.A. 1.4, Olympus, Japan) and then switched to gSTED imaging. Sections were illuminated with 488 nm light, produced by a continuous-wave laser, and short sections of non-uncaged dendrite used to optimize acquisition parameters, first under conventional confocal detection, then by gSTED imaging. The 488 nm illumination laser was set to 60–70% of maximum power, and the continuous-wave STED laser (592 nm) set to 25% and gated according to the best STED-depletion achievable in the samples (1.5–8 ms gating). Once optimized, a region of interest (ROI) was selected over the uncaged dendrite, which at 1024 × 1024 pixel size, gave a pixel resolution of 20–30 nm. Short stacks were taken over dendritic sections containing uncaged and non-spines (0.5 μm steps) with STED images interleaved with confocal images for confirmation of STED effect. STED images were deconvolved (Huygen's STED option, Scientific Volume Imaging, Netherlands) and uncaged spines identified by comparison to live 2-photon images (see Fig. 2a). Measurements of head width and neck length were then made on the deconvolved images in FIJI (ImageJ)[56].

**Serial block-face scanning-electron microscopy (SBF-SEM) of L4 SCs.** For SBF-SEM, 10 P14 mice (3 WT/7 $Fmr1^{-/y}$) were perfusion fixed. Briefly, mice were sedated with isoflurane and terminally anaesthetized with I.P. sodium pentobarbital (50 mg/mouse). The chest was opened and 10 ml of PBS (pH 7.4, filtered) transcardially perfused (~0.5 mls/second); once cleared the PBS was replaced with ice-cold fixative solution containing (3.5% PFA, 0.5% glutaraldehyde, and 15% saturated picric acid; pH 7.4), and 20 ml perfused. Brains were then removed and post-fixed overnight at 4 °C in the same fixative solution. 60 μm thick coronal sections were cut on a vibratome (Leica VT1000) and S1 identified based on visual identification. Sections were then heavy-metal substituted: first sections were rinsed in chilled PBS (5 × 3 min) and then incubated with 3% potassium ferrocyanide and 2% w/v OsO$_4$ in PBS for 1 h at 4 °C. Sections were rinsed liberally in double distilled (dd) H$_2$O and then incubated with 1% w/v thiocarbohydroxide for 20 min at room temperature. Sections were rinsed again in ddH$_2$O, and then incubated with 2% w/v OsO$_4$ for 30 min at room temperature, rinsed in ddH$_2$O and contrasted in 1% w/v uranyl acetate overnight at 4 °C. Sections were rinsed in ddH$_2$O and then contrasted with 0.6% w/v lead aspartate for 30 min at 60 °C. Sections were then rinsed in ddH$_2$O, dehydrated in serial dilutions of ethanol for 30 min each at 4 °C, then finally dehydrated twice in 100% ethanol and then 100% acetone both at 4 °C for 30 min. Sections were then impregnated with serial dilutions (25%, 50%, 75%, diluted in acetone) of Durcupan ACM (Sigma Aldrich, UK) at room temperature for 2 h per dilution, followed by 100% Durcupan ACM overnight in a dissector at room temperature. Sections were transferred to fresh Durcupan ACM for 1 h at room temperature and then flat-embedded on glass slides, coated with mould-release agent, cover-slipped, and then cured for 12 h at 60 °C.

For SFB-SEM imaging, small pieces of L4 of S1 were dissected from flat-embedded sections, with aid of a stereo microscope and glued with cyanoacrylate to stage mounting pins. The mounted tissue was then trimmed and gold-plated prior to insertion imaging. Initially, semi-thin sections trimmed from the surface of the block, and imaged under transmission electron microscopy at low power to confirm tissue ultrastructure and ROI selection for SBF-SEM. Next the tissue blocks were mounted in an SBF-SEM (3View, Gatan, CA, USA) and 3 × ~10 μm$^2$ ROIs chosen on the surface of the block, avoiding blood vessels or L4 SC somata, and imaged at 50 nm steps at ×8000 magnification (1024 × 1024, 10 nm pixel size). Approximately 100 sections were collected from each block, giving a total depth of 5 μm. SBF-SEM images were analysed offline using the TrakEM module of FIJI[57]. Dendrites and spines were traced as surface profiles and then PSDs identified on dendritic spines as electron dense regions within 25 nm of the lipid bilayer. Six to eleven dendrites were reconstructed from each mouse, which possessed a total of 38–49 spines (average = 4.4 spines/dendrite). The incidence of PSDs was calculated as an average within each mouse, and final averages produced as an animal average.

**Data analysis.** All data are presented as the mean ± SEM. Where appropriate, data were analysed with a linear (LMM) or generalised linear mixed-effects model (GLMM). Probability distributions for models were chosen by goodness of fit to normal, log-normal or gamma distributions (Figures S2 and S3). Appropriate to the particular experiment and statistical model, genotype, drug treatment and potentially their interaction were used as fixed effects, while litter, animal and slice were used as random effects. Statistical significance was assessed by likelihood ratio tests with models in which the parameter of interest had been dropped and expressed as a p-value. When animal or paired cell data are shown and not modelled, datasets were tested for normality (d'Agostino-Pearson test) and either Student's t-test, Mann–Whitney non-parametric U-test, or Wilcoxon signed-rank

tests performed. Comparison of linear and non-linear regression was performed with a sum-of-squares F-test. Statistically significant differences were assumed if $p < 0.05$. Which statistical test employed is indicated throughout the text. Either GraphPad Prism or R was used for all statistical analyses. All statistical tests performed are presented in supplementary materials (Table S1).

**Reporting summary**. Further information on research design is available in the Nature Research Reporting Summary linked to this article.

## Data availability
All datasets will be made available upon reasonable request.

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

## Acknowledgements

The authors wish to thank: Drs. Alison Dunn and Rory Duncan of the Edinburgh Super-Resolution Imaging Consortium (ESRIC) for expert advice and technical support; Kathryn Whyte and Tracey Davey of the EM Research Service, Newcastle University, for technical assistance with electron microscopy. Funders: Simons Foundation Autism Research Initiative (529085), The Patrick Wild Centre, Medical Research Council UK (MR/P006213/1), The Shirley Foundation and the RS Macdonald Charitable Trust.

## Author contributions

S.A.B.: designed and performed experiments, analysed/interpreted data and wrote the paper; A.P.F.D.: designed and interpreted, performed experiments, analysed data and wrote the paper; O.R.D.: analysed/interpreted data and wrote the paper; A.D.J.: performed experiments, analysed/interpreted data; J.T.R.I.: designed experiments and wrote the paper; G.E.H.: analysed/interpreted data, obtained funding and wrote the paper; D.J.A.W.: designed experiments, analysed/interpreted data, obtained funding and wrote the paper; P.C.K.: designed experiments, analysed/interpreted data, obtained funding and wrote the paper.

## Additional information

**Competing interests:** The authors declare no competing interests.

