## [Peer Review File · Nature Communications]

Reviewers' Comments:

Reviewer #1:

Remarks to the Author:

In this study, the authors describe cellular and circuit alterations within layer 4 of the primary somatosensory cortex at an early postnatal developmental stage in the Fmr1KO mouse model using state-of-the-art methodology ranging from electrophysiological recordings, two-photon spine glutamate uncaging, super-resolution microscopy to block-face EM. The study is well conducted and presented, and aims to tie together findings related to changes in connectivity, synaptic properties, neuronal excitability, and in dendritic properties due to a reduction in HCN current. Overall, the study provides new insights into circuit defects at an early postnatal stage (P10-15) in Fragile X Syndrome that could contribute to sensory processing deficits in this disorder. I have a few comments and suggestions for improvement or clarification and interpretation of the results.

Comments:

1. The authors use glutamate uncaging to measure dendritic integration of uEPSPs that were evoked in a near-simultaneous manner at several spines. Somatically measured responses are quantified as function of the number of spines stimulated, and the results are plotted against the expected linear sum. The authors describe this summation as linear, but the results displayed in figure 8E show a sub-linear summation for both WT and KO data, even though for KO neurons the summation is significantly enhanced. Could the authors please clarify this discrepancy/misunderstanding?
2. For the experiments measuring EPSPs following stimulation of thalamocortical projections, the authors should quantify the kinetics of the postsynaptic responses (in particular decay time constant, half-width). Judging by figure 8F it seems that in WT neurons there is a strong inhibitory component following the fast EPSPs, which is strongly reduced in KO neurons. What is the impact of this reduced inhibition on EPSP summation? Could this defect in inhibition play a major role in reducing EPSP summation, rather than the reduction in HCN channels? Have the authors repeated these experiments in the presence of GABAA receptor blockers to more directly assess the contribution of HCN channels to this process?
3. Similarly, the authors report an increase in NMDAR receptor containing synapses in Fmr1KO mice. It has previously been shown that NMDAR engagement can increase the summation of EPSPs (see Branco and Häusser, 2011). Have the authors tested the contribution of NMDAR to the increased summation in L4 SCs in Fmr1KO mice?
4. The authors should extend their analysis of the passive and active properties of neurons. This analysis should include a more thorough description of the action potential firing properties such as polarization and re-polarization rates, width, threshold, and accommodation rates. For example, in figure 6 it appears that the threshold for action potentials might be shifted in KO neurons when compared to WT neurons. Did the authors observe additional changes that are indicative of alterations in ion channels other than HCN channels? If so, which ones might be involved?
5. Did the authors measure dendritic integration only for thalamo-cortical connections or also for unitary L4-L4 connections? If so, was dendritic integration also enhanced for these cortical-cortical synapses? This knowledge would shed light on the overall impact of an HCN reduction for the integration of inputs from various brain regions. Are changes in HCN affecting mainly thalamo-cortical synapses or is the throughput of a wider range of inputs enhanced?
6. What are the overall functional consequences of the described changes in unitary L4 SC connections for the operation of the local layer 4 network (no changes in EPSC amplitude and failure rate, reduced connectivity probability, and enhanced neuronal excitability)?
7. The authors find an increased incidence of multi-innervated spines (MIS) in Fmr1KO neurons. This correlates with an increase in the uEPSC amplitude. Given that the amplitude of unitary EPSCs is unaltered and the connection probability is reduced for L4-L4 connections, the re-organization of presynaptic fibers is likely to originate somewhere outside the local L4 circuitry. Do the authors have

any idea which projections are re-organized? Could this finding be interpreted as a change in the receptive field properties of L4 SCs? Or is it more likely that different types of inputs (from different brain regions) impinge on the same spines?

8. In a couple of instances, the results are based on large differences in n-numbers between WT and KO data, and these n-numbers seem a bit low for one group each. In particular, this refers to the results for the MIS data (Figure 4; Fmr1KO mice, n = 7; WT mice, n = 3 mice), and well as for the data describing unitary EPSCs (Figure 2D-E; failure rate and unitary amplitude; Fmr1KO mice, n = 8; WT mice, n = 35).

Reviewer #2:

Remarks to the Author:

Booker et al. use several methods (glutamate uncaging, STED, serial block face EM) to investigate differences in function and morphology of single spines in layer 4 stellate cells. They report that in layer 4 SCs from Fmr1-deficient mice uEPSCs produced by glutamate uncaging onto single spines are larger (somatic recording). Rather surprisingly, this result did not carry over to unitary connections between pairs of SC neurons which were equivalent in control and Fmr1-deficient. The authors go on to demonstrate that the number of silent, NMDAR only spines increases which is presumably the underlying reason that there is an apparent reduction in connectivity between pairs of SC neurons. Using super-resolution STED imaging, there were no differences seen in spine morphology. Using EM, however, the authors discovered that there were significantly more multi-synapse spines in the Fmr1-deficient neurons. The increased number of spines with multiple inputs might explain why uEPSCs are larger but not unitary EPSCs or mEPSCs in Fmr1 lacking L4 SC neurons. The authors go on to show that the h-current (I_h) seems to be reduced in Fmr1-/-y and that summation of EPSPs more rapidly leads to spikes. Overall this is a very interesting study, the experiments have been well executed and the interpretation is in line with their results.

Major comment:

It is unclear how I_h was determined. The authors state that the current was estimated from the amplitude of the current produced in response to hyperpolarizing voltage steps. Usually I_h is determined by measuring the 'sag' current or the tail currents after returning from a strongly hyperpolarized potential to various more depolarized potentials. Alternatively, I_h could be determined from the difference in steady state currents with and without ZD-7288 (i.e. from the data in Fig. 7 panels D and E). ZD7288 is usually reported to hyperpolarize the resting membrane potential of neurons with a large I_h and to reduce the number of action potentials (unless the RMP has been re-adjusted with current injection). While the apparent reduction in I_h might be only one factor contributing to the increased dendritic integration and increased excitability of the L4 SC neurons, the authors attach a big importance to this finding so it should be solid.

Minor comments:

Page 8 line 197: the reference (Nikonenko et al., 2003) does not contain information about the % of MIS, rather this information is in several other papers from the group of the late Dominique Muller including: Toni et al., Nature 1999, Nikonenko et al., Hippocampus 2002, Nikonenko et al., J. Cell Biol. 2008.

Figure 6C: there appears something wrong with the fits – perhaps a non-linear fit would be more appropriate

Figure 8C: Why are there fewer 'N' in the Fmr1 lacking group when silent spines were excluded? If there were stretches of dendrite with all spines silent (which would explain this), then those don't appear (as 0's) in the 'all spines' red column.

page 16 line 464: Nikonenko et al., 2003 seems to be the wrong reference (see above)

Figure 7A: what are the thick and thin lines on the example traces?

Reviewer #3:

Remarks to the Author:

The authors look at individual spine structure and function in S1 L4 Stellate cells in the mouse model of FXS syndrome. They find that though structurally the spines are comparable to WT (except for the higher incidence of multi-innervated spines), functionally, individual spine responses are markedly higher in FMR1 $-/\gamma$ with an increased NMDA component. They also see a decrease in HCN current and subsequent increase in intrinsic excitability in FMR1 $-/\gamma$ neurons. The authors argue that increased single spine excitation and intrinsic excitability are the basis of cellular and circuit hyperexcitability in FXS.

Major Concerns:

1) The authors address an important question in the field of FXS: Is single spine function altered in the absence of FMRP protein. Although the authors have an exciting finding (Fig 1A-D) pointing towards increased single spine activity in FMR1 $-/\gamma$ compared to WT, the interpretation of results is confounded by the other main finding of this paper that HCN currents are markedly reduced in the absence of FMRP in S1 L4 Stellate cells.

Specifically, it would be expected that with reduced HCN currents, somatic whole-cell voltage clamp (though imperfect; refer Williams SR et al. Nat Neurosci. 2008 Jul;11(7):790-8.) would have a better space clamp, less escaped current and thus higher mEPSC amplitudes given the synaptic conductances remain the same in both conditions. The use of Cs-based internal, as the authors have done, does little to mitigate this expected HCN-based increase in EPSC amplitude as Cs is an external blocker of HCN and does not affect HCN channels internally. Thus, a central result of this study and an important one for the field remains inconclusive in spite of the exciting results in Fig. 1.

The authors should address this perhaps by doing the same uncaging experiment after a ZD block to prove conclusively that single spine responses are indeed higher in FMR1 $-/\gamma$ neurons.

2) It is concerning that the authors observed a linear increase (and not a sigmoidal or step increase) in EPSP summation even with 'near simultaneous' glutamate uncaging of multiple spines in Fig 8 when the propensity of NMDA spikes in these neurons has been well documented (Lavzin et al. 2012). Moreover, the authors themselves argue that FMR1 $-/\gamma$ neurons have a higher NMDA:AMPA ratio and enhanced NMDA component (Fig1,4). The authors should report the incidence of NMDA spikes in this data set, if present, or at least comment on the lack of it given their own findings. Ideally, these recordings should have been accompanied with Calcium imaging which seems within the technical realm of the authors and a standard in the field.

It should be noted that the authors argument of Jia et al. 2014 seeing the same 'linearity' (discussion 497-498) in an in vivo anesthetized prep with passive stimulation sounds unsatisfactory considering the work of multiple other groups showing otherwise (Lavzin 2012, Branco and Hausser 2011, Palmer LM 2012 to name a few.) in both in vivo and in vitro conditions. The authors should at least address other studies if they include the reference to Jia et al as a supporting reference.

3) The authors should clarify if 'silent synapses' were excluded in the analysis of data in Fig. 1A-D. The histogram in Fig. 1B would suggest that they were. It would be interesting to see how their inclusion

would influence the average EPSC amplitude numbers.

Minor:

1) Although the authors do an excellent job of standardizing uncaging power for their experiments (Fig S1), the calibration would mean nothing if it is not done for synapses at the same depth in the slice. The authors mention that they identified a 'superficial' spiny dendrite in the methods section but should provide a target depth or range that the uncaging data comes from.

2) The authors provide little to no details about their physiological experiments in Fig 8. for e.g What blockers were used in the voltage clamp experiment (Fig 8A) to isolate I_h ; mislabeling of data in 331-335 as mV instead of pA; characteristics of the chirp stimulus used to measure resonant characteristics; Also, the conclusion that Fmr1 $-/-$ are 'tuned to higher frequencies' is an overreach considering a jump from 0.8 to 1.1 Hz is hardly anything and more indicative of higher input resistance or 'zero impedance' considering the lack of sampling of the chirp stimulus in this frequency range.

Response to Reviewer's Comments

Reviewer #1 (Remarks to the Author):

In this study, the authors describe cellular and circuit alterations within layer 4 of the primary somatosensory cortex at an early postnatal developmental stage in the Fmr1KO mouse model using state-of-the-art methodology ranging from electrophysiological recordings, two-photon spine glutamate uncaging, super-resolution microscopy to block-face EM. The study is well conducted and presented, and aims to tie together findings related to changes in connectivity, synaptic properties, neuronal excitability, and in dendritic properties due to a reduction in HCN current. Overall, the study provides new insights into circuit defects at an early postnatal stage (P10-15) in Fragile X Syndrome that could contribute to sensory processing deficits in this disorder. I have a few comments and suggestions for improvement or clarification and interpretation of the results.

Comments:

1. The authors use glutamate uncaging to measure dendritic integration of uEPSPs that were evoked in a near-simultaneous manner at several spines. Somatically measured responses are quantified as function of the number of spines stimulated, and the results are plotted against the expected linear sum. The authors describe this summation as linear, but the results displayed in figure 8E show a sub-linear summation for both WT and KO data, even though for KO neurons the summation is significantly enhanced. Could the authors please clarify this discrepancy/misunderstanding?

We thank the reviewer for their comments. We agree that while the data fits a linear relationship the nature of the summation is sublinear relative to the line of unity. We have amended the text to clarify this point more concisely.

2. For the experiments measuring EPSPs following stimulation of thalamocortical projections, the authors should quantify the kinetics of the postsynaptic responses (in particular decay time constant, half-width). Judging by figure 8F it seems that in WT neurons there is a strong inhibitory component following the fast EPSPs, which is strongly reduced in KO neurons. What is the impact of this reduced inhibition on EPSP summation? Could this defect in inhibition play a major role in reducing EPSP summation, rather than the reduction in HCN channels? Have

the authors repeated these experiments in the presence of GABAA receptor blockers to more directly assess the contribution of HCN channels to this process?

*We agree with the reviewer that the traces shown express altered inhibitory kinetics. Indeed altered feedforward inhibition to thalamocortical stimulation in P10/11 mice is a major part of a related study (Domanski et al., BioRxiv, doi.org/10.1101/403725). To address the reviewers specific comments we have measured the kinetics of EPSPs driven by thalamocortical stimulation (**New Supplementary Figure 8**) and shows that while there is high variability in EPSP kinetics of both WT and *Fmr1^{-y}* mice, these properties are similar. However, in Domanski et al., manuscript, we demonstrate an increase in feedforward inhibition onto the layer 4 spiny stellate neurons; however, this inhibition is delayed resulting in an increase in the half-width of the EPSP. Thus, together our data demonstrate that both reduction in the overall function current through HCN channels as well as mistimed inhibition contribute to the altered summation.*

3. Similarly, the authors report an increase in NMDAR receptor containing synapses in *Fmr1*KO mice. It has previously been shown that NMDAR engagement can increase the summation of EPSPs (see Branco and Häusser, 2011). Have the authors tested the contribution of NMDAR to the increased summation in L4 SCs in *Fmr1*KO mice?

*We have now included new data showing that EPSP summation in L4 SCs dendrites is strongly attenuated by the NMDA receptor antagonist AP-5 selectively in *Fmr1^{-y}* mice (**new Supplementary Figure 7**). This is in agreement with the findings that NMDA receptor function is increased at single dendritic spines, likely due to increased multiple-innervation of spines in the *Fmr1^{-y}* mouse.*

4. The authors should extend their analysis of the passive and active properties of neurons. This analysis should include a more thorough description of the action potential firing properties such as polarization and re-polarization rates, width, threshold, and accommodation rates. For example, in figure 6 it appears that the threshold for action potentials might be shifted in KO neurons when compared to WT neurons. Did the authors observe additional changes that are indicative of alterations in ion channels other than HCN channels? If so, which ones might be

involved?

*We have extended our analysis of passive and active properties to address their concerns. We find that no other major intrinsic electrophysiological property was altered over the age range we recorded. A new **Supplementary Figure 4** has been included to summarise these findings. Furthermore, we have performed additional experiments assessing the function of HCN channels in L4 SCs (new **Figure 8**), which confirms that I_h activation is functionally reduced in $Fmr1^{-/y}$ mice, through a cyclic-AMP dependent mechanism.*

5. Did the authors measure dendritic integration only for thalamo-cortical connections or also for unitary L4-L4 connections? If so, was dendritic integration also enhanced for these cortical-cortical synapses? This knowledge would shed light on the overall impact of an HCN reduction for the integration of inputs from various brain regions. Are changes in HCN affecting mainly thalamo-cortical synapses or is the throughput of a wider range of inputs enhanced?

The reviewer raises an interesting point that is not trivial to address experimentally. Given large reduction in paired connectivity between layer 4 SCs (Figure 2C), the altered NMDA/AMPA content and the increase in silent synapses, which are likely to be cortico-cortical in nature at this age (Harlow et al., 2011), a direct comparison of dendritic integration between genotypes would not be easily interpreted. Therefore, directly testing the summation of cortical-cortical synapses would require the selective identification of these synapses combined with glutamate uncaging and is beyond the scope of the current study. Furthermore, it would not directly alter our interpretation of the findings reported in this study.

6. What are the overall functional consequences of the described changes in unitary L4 SC connections for the operation of the local layer 4 network (no changes in EPSC amplitude and failure rate, reduced connectivity probability, and enhanced neuronal excitability)?

Circuit hyperexcitability within somatosensory cortex has been well characterised in vitro and in vivo (Gibson et al 2008, Bureau et al., 2008, Harlow et al., 2011, Zhang et al., 2014). The present study explores the synaptic/cellular and subcellular mechanisms for local circuit hyperexcitability. Indeed, we show increased cellular excitability through reduced I_h , increased silent spines and increased multi-innervated spines, which converge to provide such a cellular substrate for the circuit alterations. Furthermore, we provide new data that shows an increase

*frequency of spontaneous EPSC in L4 SCs in the FXS mouse model, consistent with increased local circuit activity. This data has been included in the new **Supplementary Figure S5** and described in the results. Finally, the functional output of layer 4 to layer 2/3 is explored in Domanski et al. [BioRxiv, doi.org/10.1101/403725](https://doi.org/10.1101/403725)*

7. The authors find an increased incidence of multi-innervated spines (MIS) in Fmr1KO neurons. This correlates with an increase in the uEPSC amplitude. Given that the amplitude of unitary EPSCs is unaltered and the connection probability is reduced for L4-L4 connections, the re-organization of presynaptic fibers is likely to originate somewhere outside the local L4 circuitry. Do the authors have any idea which projections are re-organized? Could this finding be interpreted as a change in the receptive field properties of L4 SCs? Or is it more likely that different types of inputs (from different brain regions) impinge on the same spines?

We hypothesized that the increased synapse density arises from both local L4 and TCA afferents. This hypothesis is supported by previous work from Wang et al. (2014) showing an increase in both vGluT1 and vGluT2-containing axon terminals. As noted in our discussion, the findings of Harlow et al., strongly suggest that silent synapses are largely absent at TCA inputs at the age examined in this study. Therefore, the increase in silent synapses seen in this study are likely to be at cortical-cortical synapses. Indeed these data highlight a progressive delay in cortical synaptic development, first starting at TCA synapses (Harlow et al., 2011), followed by cortico-cortical synapses in layer 4 (current manuscript). It is notable that the lack of any change in mEPSCs in the Fmr1^{-y} animals strongly suggests a normal complement of synapses with AMPA receptors. How this relates to the MIS is not known, however, the increase in vGluT1 containing terminals suggest it may result from an increase in TCA terminals. Similarly, how the altered synaptogenesis relates to receptive field properties is not known. On this note, it is important to remember that this is an early stage of development when experience is still shaping receptive field properties. How the altered patterns of activity in layer 4 may be affecting the development of activity patterns in layer 2/3 cells is a focus of the Domanski et al manuscript. While these questions are all interesting, they are well beyond the scope of the current manuscript which is focussed on how cellular changes in neurons lacking FMRP alter the synaptic and dendritic integration properties during a developmental time-window when firing patterns are key to mediating experience-dependent shaping of somatosensory cortex.

8. In a couple of instances, the results are based on large differences in n-numbers between WT and KO data, and these n-numbers seem a bit low for one group each. In particular, this refers to the results for the MIS data (Figure 4; Fmr1KO mice, n = 7; WT mice, n = 3 mice), and well as for the data describing unitary EPSCs (Figure 2D-E; failure rate and unitary amplitude; Fmr1KO mice, n = 8; WT mice, n = 35).

*The differences in n are a result of practical issues when using mice at an age when they cannot be genotyped prior to the experiment. As such, there is an inherent randomness in the number of mice per genotype collected for each experiment. To attempt to overcome this we collected more biological replicates than initial power analysis suggested, however, in some instances the sample size from each genotype was not equal. That said, the use of linear mixed models largely overcomes statistical issues that occur with uneven sample sizes across experimental groups. This data is presented in a new **Supplemental Table 1**.*

With reference to the specific examples stated, we performed electron microscopic analysis of 2x litters of male mice (n=12 mice total) which resulted in the numbers obtained. Each value (N) represents 5-10 reconstructed dendrites and approximately 50 spines per mouse. After two mice were excluded prior to analysis, due to very poor ultrastructure and/or lead citrate contrasting, we had only 3 WT (n=3 mice) for analysis. It is important to note, however, that the percentage of MIS obtained in our study are closely matched to those reported in previous studies from similarly aged animals, albeit in organotypic slice culture (Nikonenko et al., 2008). Thus, the substantial increase in MIS' in Fmr1^{-y} animals is not only in comparison to WT animals in this study, but also in comparison to previous literature.

With respect to paired recordings and the unitary EPSC data the lower number of Fmr1^{-y} neurons is partially due to relatively fewer Fmr1^{-y} mice in the litters tested (see above), however, this is exacerbated by the lower connection probability of Fmr1^{-y} mice. However, as noted above, the use of LMM for statistical analysis overcomes this imbalance in n between experimental groups.

Reviewer #2 (Remarks to the Author):

Booker et al. use several methods (glutamate uncaging, STED, serial block face EM) to

investigate differences in function and morphology of single spines in layer 4 stellate cells. They report that in layer 4 SCs from Fmr1-deficient mice uEPSCs produced by glutamate uncaging onto single spines are larger (somatic recording). Rather surprisingly, this result did not carry over to unitary connections between pairs of SC neurons which were equivalent in control and Fmr1-deficient. The authors go on to demonstrate that the number of silent, NMDAR only spines increases which is presumably the underlying reason that there is an apparent reduction in connectivity between pairs of SC neurons. Using super-resolution STED imaging, there were no differences seen in spine morphology. Using EM, however, the authors discovered that there were significantly more multi-synapse spines in the Fmr1-deficient neurons. The increased number of spines with multiple inputs might explain why uEPSCs are larger but not unitary EPSCs or mEPSCs in Fmr1 lacking L4 SC neurons. The authors go on to show that the h-current (I_h) seems to be reduced in Fmr1-/-y and that summation of EPSPs more rapidly leads to spikes. Overall this is a very interesting study, the experiments have been well executed and the interpretation is in line with their results.

Major comment:

It is unclear how I_h was determined. The authors state that the current was estimated from the amplitude of the current produced in response to hyperpolarizing voltage steps. Usually I_h is determined by measuring the 'sag' current or the tail currents after returning from a strongly hyperpolarized potential to various more depolarized potentials. Alternatively, I_h could be determined from the difference in steady state currents with and without ZD-7288 (i.e. from the data in Fig. 7 panels D and E). ZD7288 is usually reported to hyperpolarize the resting membrane potential of neurons with a large I_h and to reduce the number of action potentials (unless the RMP has been re-adjusted with current injection). While the apparent reduction in I_h might be only one factor contributing to the increased dendritic integration and increased excitability of the L4 SC neurons, the authors attach a big importance to this finding so it should be solid.

*We thank the reviewer for their helpful comments and we have conducted their suggested experiments as well as further experiments to directly test I_h . First, we show that both voltage "sag" and rebound slope (as performed in Brager et al., 2012) are reduced in L4 SCs from Fmr1^{-/-y} mice, indicating reduced I_h at these membrane potentials (**Revised Figure 7**). Furthermore, we performed additional voltage-clamp recordings in which we pharmacologically isolated I_h (**New Figure 8**). Intriguingly, these data demonstrate that I_h displays a shifted*

*activation, rather than reduced peak current. To understand the mechanism of this shifted activation, we assessed the sensitivity of I_h to the adenylyl cyclase activator, forskolin. As shown in new **Figure 8**, forskolin normalises the I_h activation between WT and $Fmr1^{-/y}$ L4 SCs. This finding is consistent with the notion that there is diminished neuronal cyclic-AMP signalling in the mouse model of Fragile X syndrome, as previously reported (Berry-Kravis et al., 1995, Kelley et al., 2007, 2008; Choi et al., 2016). Indeed, our data imply that rather than altered HCN channel expression (Brager et al., 2014; Zhang et al., 2014) it is the reduction in cyclic-AMP that leads to a reduction in HCN-channel currents observed at physiologically relevant membrane potentials. Further comment of this new data has been included in the discussion.*

Minor comments:

Page 8 line 197: the reference (Nikonenko et al., 2003) does not contain information about the % of MIS, rather this information is in several other papers from the group of the late Dominique Muller including: Toni et al., Nature 1999, Nikonenko et al., Hippocampus 2002, Nikonenko et al., J. Cell Biol. 2008.

We have now added the reference to Nikonenko et al., 2008 to this section. Given that Toni et al., Nature 1999, Nikonenko et al., 2002 Hippocampus, and Nikonenko 2003 J. Neurosci, deal with presynaptic and activity dependent aspects of spinogenesis, we have instead included these references in the discussion, as they deal with perforated synapses and multi-spine boutons, rather than multi-bouton spines.

Figure 6C: there appears something wrong with the fits – perhaps a non-linear fit would be more appropriate

We thank the reviewer for their comment, we had fitted a straight line to the linear phase of the IV plot. However, as we test the data with a 2-way ANOVA we have now presented the data as contiguous points.

Figure 8C: Why are there fewer 'N' in the $Fmr1$ lacking group when silent spines were excluded? If there were stretches of dendrite with all spines silent (which would explain this), then those don't appear (as 0's) in the 'all spines' red column.

This discrepancy arises from a handful of cells (n=5) for which the summing EPSP to multiple spine activation was examined, which was sufficient to report the number of spines required to reach threshold, but for which the EPSP for individual spines were not measured. As such we do not know whether these dendrites contained "silent" synapses and they were excluded from this analysis. These cells were used included only in figure 8B, but not in any other analysis. A line has been added to this effect.

page 16 line 464: Nikonenko et al., 2003 seems to be the wrong reference (see above)

We have now expanded this discussion point to incorporate the Muller laboratory's body of work on this subject.

Figure 7A: what are the thick and thin lines on the example traces?

As mentioned above, we have restructured figures 7 and 8 to address the reviewers' comments. These traces have been replaced in the new figures with data that directly examines I_h .

Reviewer #3 (Remarks to the Author):

The authors look at individual spine structure and function in S1 L4 Stellate cells in the mouse model of FXS syndrome. They find that though structurally the spines are comparable to WT (except for the higher incidence of multi-innervated spines), functionally, individual spine responses are markedly higher in FMR1 $-/-$ with an increased NMDA component. They also see a decrease in HCN current and subsequent increase in intrinsic excitability in FMR1 $-/-$ neurons. The authors argue that increased single spine excitation and intrinsic excitability are the basis of cellular and circuit hyperexcitability in FXS.

Major Concerns:

1) The authors address an important question in the field of FXS: Is single spine function altered in the absence of FMRP protein. Although the authors have an exciting finding (Fig 1A-D)

pointing towards increased single spine activity in FMR1 ^{-/-} compared to WT, the interpretation of results is confounded by the other main finding of this paper that HCN currents are markedly reduced in the absence of FMRP in S1 L4 Stellate cells.

Specifically, it would be expected that with reduced HCN currents, somatic whole-cell voltage clamp (though imperfect; refer Williams SR et al. Nat Neurosci. 2008 Jul;11(7):790-8.) would have a better space clamp, less escaped current and thus higher mEPSC amplitudes given the synaptic conductances remain the same in both conditions. The use of Cs-based internal, as the authors have done, does little to mitigate this expected HCN-based increase in EPSC amplitude as Cs is an external blocker of HCN and does not affect HCN channels internally. Thus, a central result of this study and an important one for the field remains inconclusive in spite of the exciting results in Fig. 1.

The authors should address this perhaps by doing the same uncaging experiment after a ZD block to prove conclusively that single spine responses are indeed higher in FMR1 ^{-/-} neurons.

We thank the reviewer for their comment. It is important to remember that SCs are small neurons with compact dendrites, especially at this age and therefore much better space clamp can be achieved, relative to older neurons or large, pyramidal cortical neurons. To directly test this, we examined uEPSC rise, decay and amplitudes as a function of spine distance from the cell soma in WT mice, where the increase in HCN currents would be expected to have a greater effect on space clamp. We see no difference in rise or decay time as a function of spine distance from cell soma. Furthermore, we see an increase in uEPSC amplitude with distance from cell soma, the opposite of what would be expected with poor space clamp. This data is now shown in supplemental Figure 1F-H. In addition, to address directly the reviewer's concern we performed recordings from WT and Fmr1^{-/-} mice to assess the effect of I_h block on spontaneous EPSC amplitude and kinetics. We observed no difference between genotypes in EPSC amplitude, frequency or kinetics following ZD application, indicating that under our recording conditions there was no genotypic difference in space clamp. The results of these additional experiments have been included in new Supplementary Figure 5.

2) It is concerning that the authors observed a linear increase (and not a sigmoidal or step increase) in EPSP summation even with 'near simultaneous' glutamate uncaging of multiple

spines in Fig 8 when the propensity of NMDA spikes in these neurons has been well documented (Lavzin et al. 2012). Moreover, the authors themselves argue that FMR1 ^{-/-} neurons have a higher NMDA:AMPA ratio and enhanced NMDA component (Fig1,4). The authors should report the incidence of NMDA spikes in this data set, if present, or at least comment on the lack of it given their own findings. Ideally, these recordings should have been accompanied with Calcium imaging which seems within the technical realm of the authors and a standard in the field.

We thank the reviewer for their comment. There are many experimental differences between our study and that of Lavzin et al. (2012) including method of stimulation and the age of animals. Indeed, they used an elegant study design to specifically drive NMDAR spikes, something we have not done in our study. Under our recording conditions, we found no evidence for NMDA spikes in either genotype and hence they are unlikely to contribute to our results and do not affect our main conclusions that synaptic integration resulting from altered I_h is aberrant in layer 4 SCs in the absence of FMRP.

It should be noted that the authors argument of Jia et al. 2014 seeing the same ‘linearity’ (discussion 497-498) in an in vivo anesthetized prep with passive stimulation sounds unsatisfactory considering the work of multiple other groups showing otherwise (Lavzin 2012, Branco and Hausser 2011, Palmer LM 2012 to name a few.) in both in vivo and in vitro conditions. The authors should at least address other studies if they include the reference to Jia et al as a supporting reference.

We agree that our discussion of the sub-linearity of summation was not fully discussed in the context of the existing body of literature. As such we have extended our discussion of this and, with respect to the above comment, discuss the absence of non-linearity in more detail.

3) The authors should clarify if ‘silent synapses’ were excluded in the analysis of data in Fig. 1A-D. The histogram in Fig. 1B would suggest that they were. It would be interesting to see how their inclusion would influence the average EPSC amplitude numbers.

In figure 1B-D we only include non-silent spines, as we assessed the single-spine AMPA currents produced. We include one example “silent spine” trace in Panel 1A to illustrate the variability of the uEPSCs we observed. We have added a line to this effect in the figure legend.

Notwithstanding, we still observed a statistically significant difference between genotypes when “silent spines” were included (WT: 6.5 ± 0.4 pA, Fmr1^{-/-}: 8.1 ± 0.5 pA, $P=0.002$, t-test), indicating that the increase in silent spines does not “average out” the increase in amplitude of the AMPA-containing spines to create and equal overall input to the neurons.

Minor:

1) Although the authors do an excellent job of standardizing uncaging power for their experiments (Fig S1), the calibration would mean nothing if it is not done for synapses at the same depth in the slice. The authors mention that they identified a ‘superficial’ spiny dendrite in the methods section but should provide a target depth or range that the uncaging data comes from.

We have now added a line to the methods.

2) The authors provide little to no details about their physiological experiments in Fig 8. for e.g What blockers were used in the voltage clamp experiment (Fig 8A) to isolate I_h; mislabeling of data in 331-335 as mV instead of pA; characteristics of the chirp stimulus used to measure resonant characteristics; Also, the conclusion that Fmr1^{-/-} are ‘tuned to higher frequencies’ is an overreach considering a jump from 0.8 to 1.1 Hz is hardly anything and more indicative of higher input resistance or ‘zero impedance’ considering the lack of sampling of the chirp stimulus in this frequency range.

*The addition of extensive new experiments strengthening the role I_h in L4 SCs has resulted in a new section of the results fully describing the pharmacology employed, as well as changes to **Figure 7** and a new **Figure 8**. We hope that sufficient detail is now provided. The mistake on lines 331-335 has been rectified. We have been toned down our interpretation of minor change in resonant frequency.*

Reviewers' Comments:

Reviewer #1:

Remarks to the Author:

Previous question: Did the authors measure dendritic integration only for thalamo-cortical connections or also for unitary L4-L4 connections? If so, was dendritic integration also enhanced for these cortical-cortical synapses? This knowledge would shed light on the overall impact of an HCN reduction for the integration of inputs from various brain regions. Are changes in HCN affecting mainly thalamo-cortical synapses or is the throughput of a wider range of inputs enhanced?

Response: The reviewer raises an interesting point that is not trivial to address experimentally. Given large reduction in paired connectivity between layer 4 SCs (Figure 2C), the altered NMDA/AMPA content and the increase in silent synapses, which are likely to be cortico-cortical in nature at this age (Harlow et al., 2011), a direct comparison of dendritic integration between genotypes would not be easily interpreted. Therefore, directly testing the summation of cortical-cortical synapses would require the selective identification of these synapses combined with glutamate uncaging and is beyond the scope of the current study. Furthermore, it would not directly alter our interpretation of the findings reported in this study.

Perhaps the question was not clearly posed. Since the authors are performing simultaneous recordings from synaptically connected L4 neuron pairs, they are in the position to measure the integration of these unitary synaptic responses to trains of presynaptic action potentials. This should enable them to compare synaptic integration for unitary L4-L4 connections between genotypes, and also with that of thalamo-cortical connections.

Choice of traces: The example shown in Figure 7H displays a very strong effect of ZD on the voltage responses, yet according to the group data (Fig. 7E) the effect of ZD on the input resistance is not significant in *Fmr1*^{-/y} neurons. The authors might wish to select a more representative example for this figure.

Reviewer #2:

Remarks to the Author:

Booker et al. study the mechanisms of hyperexcitability in *FMRP*^{-/y} mice. Combining different methods, they convincingly show that a combination of aberrant spine innervation, enhanced NMDA currents and reduced *I_h* leads to the enhanced excitability of layer 4 stellate cells in the absence of *FMRP*. This is an important study, showing for the first time that *FMRP* has very strong effects on dendritic integration, increases the number of silent synapses and reduces connectivity in layer 4. The latter effects might reflect the system's attempts to restore homeostasis. In the revised version, the authors provide a convincing molecular mechanism for the enhanced dendritic summation in KO mice, namely deficient cAMP signaling that leads to a significant shift in *I_h* activation threshold. These mechanistic insights may open new avenues to combat FXS during early development.

In summary, the study provides an integrated view how the lack of *FMRP* causes hyperexcitability of stellate cells, the first stage of cortical processing, and adds a cautionary note with regard to light microscopic analysis of spine morphology, a commonly used method that encourages speculation but does not provide reliable information about synaptic function. All claims are solid, backed up by precise measurements and appropriate statistics, previous studies are appropriately cited.

The only point that is still not clearly stated in the manuscript is the exclusion of silent spines from the

analysis in Fig. 1B-D (see Reviewer 3, major point 3). Contrary to the author's rebuttal, this information is still lacking from the figure legend. The statistical analysis of ALL uncaged spines (showing a significant difference according to the author's response) should also be included in the manuscript, as the reader otherwise wonders whether the stronger synapses in the KO are fully compensated by the larger fraction of silent spines (the answer is: no). Otherwise, I have no further comments or criticism.

Reviewer #3:

Remarks to the Author:

In reference to authors reply to point. 3-1 of the reviewer's comments:

The authors provide data in Supplemental fig.5 to support their argument that differences in I_h do not influence the quality of their space clamp for the two genotypes. They state:

'We observed no difference between genotypes in EPSC amplitude, frequency or kinetics following ZD application, indicating that under our recording conditions there was no genotypic difference in space clamp.'

The results shown are intriguing as although there are no differences before and after ZD between genotypes as the authors argue, there also seems no difference in amplitude of spontaneous EPSCs between the two genotypes in the control condition or after ZD (Fig.S5e). This seems in stark contrast to the main finding of the paper and Fig.1 depicting differences in unitary EPSC amplitudes. The authors should address this discrepancy in their results between elicited unitary and spontaneous EPSC differences between the two genotypes.

The authors have adequately addressed all my other concerns.

Booker et al., response to reviewer's comments:

We would like to thank the reviewers again for their thoughtful comments. Below the reviewer's comments are highlighted in bold followed by our responses.

Reviewer #1 (Remarks to the Author):

In the first round of reviews, reviewer 1 asked,

"Did the authors measure dendritic integration only for thalamo-cortical connections or also for unitary L4-L4 connections? If so, was dendritic integration also enhanced for these cortical-cortical synapses? This knowledge would shed light on the overall impact of an HCN reduction for the integration of inputs from various brain regions. Are changes in HCN affecting mainly thalamo-cortical synapses or is the throughput of a wider range of inputs enhanced?"

In the second round of the review process, reviewer 1 stated,

" Perhaps the question was not clearly posed. Since the authors are performing simultaneous recordings from synaptically connected L4 neuron pairs, they are in the position to measure the integration of these unitary synaptic responses to trains of presynaptic action potentials. This should enable them to compare synaptic integration for unitary L4-L4 connections between genotypes, and also with that of thalamo-cortical connections."

We believe there is still some confusion. While we are looking at paired connectivity, we have no way of knowing the number of synapses between connected pairs, nor control for the location of the synapses on the dendrites. As such, there would be many different interpretations of the data beyond the impact of HCN channels. Furthermore, the relatively low connection probability between layer 4 SCs in *Fmr1*^{-y} animals means that these experiments would be very time consuming to achieve a result with multiple interpretations. Instead, we plan to directly test the summation of cortical-cortical synapses, which we show are the likely source of the silent synapses, through selective identification of these synapses combined with glutamate uncaging. However, this constitutes a large study that is well-beyond the scope of the current study. Furthermore, it would not directly alter our interpretation of the findings reported in this study.

Choice of traces: The example shown in Figure 7H displays a very strong effect of ZD on the voltage responses, yet according to the group data (Fig. 7E) the effect of ZD on the input resistance is not significant in *Fmr1*^{-y} neurons. The authors might wish to select a more representative example for this figure.

The reviewer is correct. The trace shown was not representative and has now been replaced with a more representative example.

Reviewer #2 (Remarks to the Author):

The only point that is still not clearly stated in the manuscript is the exclusion of silent spines from the analysis in Fig. 1B-D (see Reviewer 3, major point 3). Contrary to the author's rebuttal, this

information is still lacking from the figure legend. The statistical analysis of ALL uncaged spines (showing a significant difference according to the author's response) should also be included in the manuscript, as the reader otherwise wonders whether the stronger synapses in the KO are fully compensated by the larger fraction of silent spines (the answer is: no). Otherwise, I have no further comments or criticism.

We apologise for this oversight. This has now been added to the figure legend. However, adding analysis that includes those spines with no response using a generalised linear mixed model is not straightforward or statistically correct. The uEPSP amplitude shown in figure 1D **does not** exclude silent spines, and as such reflects the compensation of synaptic strength. It demonstrates that the genotypic differences remain even when the silent spines are included in the analysis.

Reviewer #3 (Remarks to the Author):

In reference to authors reply to point. 3-1 of the reviewer's comments:

The authors provide data in Supplemental fig.5 to support their argument that differences in Ih do not influence the quality of their space clamp for the two genotypes. They state:

'We observed no difference between genotypes in EPSC amplitude, frequency or kinetics following ZD application, indicating that under our recording conditions there was no genotypic difference in space clamp.'

The results shown are intriguing as although there are no differences before and after ZD between genotypes as the authors argue, there also seems no difference in amplitude of spontaneous EPSCs between the two genotypes in the control condition or after ZD (Fig.S5e). This seems in stark contrast to the main finding of the paper and Fig.1 depicting differences in unitary EPSC amplitudes. The authors should address this discrepancy in their results between elicited unitary and spontaneous EPSC differences between the two genotypes.

The authors have adequately addressed all my other concerns.

We believe there is some confusion between the uncaged EPSCs (uEPSCs) showed in Figure 1 and the spontaneous EPSCs (sEPSCs) shown in Fig. S5e. The uEPSCs reflect both single-innervated and multi-innervated spines. Because of the latter, which are far more prevalent in *Fmr1*^{-/-}, the mean uEPSC is greater in the *Fmr1*^{-/-} mice. In contrast the sEPSCs are action potential driven and even in the case of multi-innervated spines, only one synapse will be activated. In support of this interpretation the amplitude of the mEPSCs (Figure 5), unitary EPSCs (Figure 2) and sEPSCs are comparable. Hence, at this age sEPSCs likely represent the activation of a single synapse. We have added clarification for this point (line 500).

Reviewers' Comments:

Reviewer #1:

Remarks to the Author:

I have no further comments/suggestions.

Andreas Frick

Reviewer #2:

Remarks to the Author:

I have no further concerns.

Reviewer #3:

Remarks to the Author:

The authors have addressed my concern from the first round of reviews satisfactorily and I recommend the manuscript for publication.